# Geothermal heat flux from measured temperature profiles in deep ice boreholes in Antarctica

Pavel Talalay[1], Yazhou Li[1], Laurent Augustin[2], Gary D. Clow[3], Jialin Hong[1], Eric Lefebvre[4], Alexey Markov[1], Hideaki Matoyama[5], Catherine Ritz[4]

[1]Polar Research Center, Institute for Polar Science and Engineering, Jilin University, 130021 Changchun, China
[2]Division Technique de l'INSU, CNRS, 83507 La Seyne sur Mer, France
[3]Institute of Arctic and Alpine Research, University of Colorado Boulder, Boulder, Colorado, USA
[4]Université Grenoble Alpes, CNRS, IRD, IGE, 38000 Grenoble, France
[5]National Institute of Polar Research, Tokyo, Japan

*Correspondence to:* Pavel Talalay (ptalalay@yahoo.com) and Yazhou Li (jluyazhouli@163.com)

**Abstract**. The temperature at the Antarctic ice sheet bed and the temperature gradient in subglacial rocks have been directly measured only a few times, although extensive thermodynamic modeling has been used to estimate the geothermal heat flux (GHF) under the ice sheet. During the last five decades, deep ice-core drilling projects at six sites – Byrd, WAIS Divide, Dome C, Kohnen, Dome F, and Vostok – have succeeded in reaching to, or nearly to, the bed at inland locations in Antarctica. When temperature profiles in these boreholes and steady-state heat flow modeling are combined with estimates of vertical velocity, the heat flow at the ice-sheet base is translated to a geothermal heat flux of 57.9±6.4 mW m$^{-2}$ at Dome C, 78.9±5.0 mW m$^{-2}$ at Dome F, and 86.9±16.6 mW m$^{-2}$ at Kohnen, all higher than the predicted values at these sites. This warm base under the East Antarctic Ice Sheet (EAIS) could be caused by radiogenic heat effects or hydrothermal circulation not accounted for by the models. The GHF at the base of the ice sheet at Vostok has a negative value of -3.6±5.3 mW m$^{-2}$, indicating that water from Lake Vostok is freezing onto the ice sheet base. Correlation analyses between modeled and measured depth-age scales at the EAIS sites indicate that all of them can be adequately approximated by a steady-state model. Horizontal velocities and their variation over ice-age cycles are much greater for the West Antarctic Ice Sheet than for the interior EAIS sites; a steady-state model cannot precisely describe the temperature distribution here. Even if the correlation factors for the best fitting age-depth curve are only moderate for the West Antarctic sites, the GHF values estimated here 88.4±7.6 mW m$^{-2}$ at Byrd and 113.3±16.9 mW m$^{-2}$ at WAIS Divide can be used as references before more precise estimates are made on the subject.

## 1 Introduction

The Antarctic geothermal heat flux (GHF), an important boundary condition for ice sheet behavior, can influence sea-level changes (Golledge et al., 2015) considering its significant influence on the viscosity of basal ice and meltwater content at the ice-base interface. What are the basal ice temperature and mechanical properties? How does GHF control basal melt and affect the internal deformation of the ice sheet? How old is ice at different locations? These questions can be answered only by applying reliable GHF measurements or estimates.

The average global surface GHF is ~86 mW m$^{-2}$, which varies from 64.7 mW m$^{-2}$, the mean continental heat flow (including arcs and continental margins), to 95.9 mW m$^{-2}$, the mean oceanic heat flow (Davies, 2013). However, several geologic factors including heat from the mantle, heat production in the crust by radioactive decay, and tectonic history, cause spatially variable
GHF in Antarctica (Burton-Johnson et al., 2020).

Most studies of GHF in Antarctica rely on thermal models (Pattyn, 2010; Van Liefferinge and Pattyn, 2013). Modeling studies based on a global seismic survey and the structural similarity of crust and upper mantle showed that the West Antarctic Ice Sheet (WAIS) has a GHF three times higher than that of the East Antarctic ice sheet (EAIS) (Shapiro and Ritzwoller, 2004). For a central point in the WAIS (78S, 110W), the average GHF is expected to be 110 mW m$^{-2}$. The GHF can also be estimated
on the basis of geologic information, where uniform values are attributed to large geologically homogeneous areas (An et al., 2015; Goodge, 2018; Llubes et al., 2006; Martos et al., 2015; Pollard et al., 2005).

Some studies use remote methods to estimate the GHF underneath the Antarctic ice sheet. For example, satellite magnetic data showed that the GHF underneath the ice sheet varies from 40 to 185 mW m$^{-2}$ and that areas of high GHF coincide with known current volcanism and some areas known to have ice streams (Fox Maule et al., 2005). In the central part of the EAIS, the
average GHF was estimated to be in the range of 50 to 60 mW m$^{-2}$; however, elevated GHFs were found along the WAIS– EAIS boundary and around the Siple Coast. Similarly, high GHFs were found around Victoria Land, Oates Land, and George V Land. Observations of crustal heat production within the continental crust underneath the Lambert-Amery glacial system in East Antarctica also show high heat flux of at least 120 mW m$^{-2}$ (Pittard et al., 2016).

Direct temperature measurement obviously produces the most reliable GHF estimates and can be used to verify results of
preliminary thermal modeling and geological-geophysical studies. While over 10,000 heat flow measurements have been made globally, 90% are from Europe, North America, and southern Africa. South America, Asia, and Australia have far fewer measurements, while Antarctica has virtually none (Davies, 2013). Drilling through thick ice is extremely complicated, time-consuming, and expensive; therefore, direct temperature measurements in Antarctic subglacial till/bedrock environments have only been conducted twice so far, both under the WAIS: at the subglacial Lake Whillans (285±80 mW m$^{-2}$) (Fisher et al., 2015)
and near the grounding zone of the Whillans Ice Stream (88±7 mW m$^{-2}$) (Begeman et al., 2017), ~100 km apart (Fig. 1). The tremendous difference in the values of GHF between these two adjacent sites suggests high spatial variability in West Antarctica.

More reliable GHF estimates under the Antarctic ice sheet can be made from available temperature profiles in ice boreholes. During the last five decades, deep ice-core drilling projects at six sites – Byrd (Ueda and Garfield, 1970), WAIS Divide
(Slawny et al., 2014), Dome C (Augustin et al., 2007), Kohnen (Wilhelms et al., 2014), Dome F (Motoyama, 2007), and Vostok (Lukin and Vasiliev, 2014; Vasiliev et al., 2011) – have succeeded in reaching, or nearly reaching, the ice sheet bed at inland locations in Antarctica. Reported drill site conditions – snow accumulation rate, mean annual surface air temperature, ice sheet surface velocity, ice thickness, and drilling depth – are summarized in Table 1.

The Byrd and Kohnen holes encountered water at the base, which welled up into the holes. The borehole at Vostok penetrated
the subglacial Lake Vostok at 3769.3 m, and here, water rose from the lake to a height of more than 340 m. Drilling of the

other holes was stopped within 10–50 m of the bed. All these holes were temperature logged and provide a good opportunity to fill the gap in our knowledge of the GHF under the Antarctic ice.

## 2 Methods

### 2.1 Temperature and temperature gradient at the base of Antarctic Ice Sheet

Temperatures in the Byrd, WAIS Divide, Vostok, Dome C, Kohnen, and Dome F boreholes were measured using different devices and different methods. All boreholes were mechanically drilled and filled with kerosene-based drilling fluid. Temperature profiles were then obtained by logging with custom-made borehole loggers (Dome C, Kohnen, Dome F, WAIS Divide, and Vostok) or a thermistor embedded in the drill (Byrd).

Measured temperature profiles in four of the boreholes (Vostok, Dome C, Kohnen, and Dome F) increase almost linearly with
depth, as expected at locations with minimal annual snow accumulation and hence small vertical velocities (Fig. 2). In contrast, vertical advection is much greater at the Byrd and WAIS Divide sites in West Antarctica; at these locations the upper part of the ice sheet is nearly isothermal, but at depth the temperature gradient is nearly the same as that at the other sites. Temperature gradients at the bed are 2.02–3.12 °C/100 m at Dome C, Kohnen, Dome F and Vostok and slightly higher in West Antarctica, 3.70–3.88 °C/100 m at Byrd and WAIS Divide (Table 2).

Temperature profiles in deep ice-core drilling boreholes are approximated closely by polynomials with correlation factors of >0.99 (Table 3), indicating a positive relationship between temperature and vertical depth. Ice thicknesses generated by extrapolating the temperature profile to the depth of the pressure melting point assuming a Clausius-Clapeyron slope of 0.0742 K/MPa (Cuffey and Paterson, 2010) are in good agreement with radar data, except for WAIS Divide where the difference is ~30 m. This could be attributed to scintillations on the melted ice-bedrock interface or other effects. However, in-depth
temperature extrapolation has limited accuracy and thus often does not provide a correct estimate of GHF. Thus, a steady-state model and genetic algorithm (GA) are applied herein to fit the measured temperatures.

### 2.2 GHF estimation model

A one-dimensional time-dependent energy-balance equation (Dahl-Jensen et al., 2003; Johnsen et al., 1995) is usually used to model the temperature distribution through the ice as a function of the climate conditions on the surface and the GHF from the
bedrock:

$$\rho c \frac{\partial T}{\partial t} = \frac{\partial}{\partial z}\left(k \frac{\partial T}{\partial z}\right) - \rho c w \frac{\partial T}{\partial z} - \rho c u \frac{\partial T}{\partial x}, \tag{1}$$

where $T$ is the temperature as a function of $z$, $x$ and $t$; $z$ represents the vertical coordinate (at the ice sheet base, $z = 0$, while at ice sheet surface, $z = H$); $H$ is the ice thickness and is assumed to be constant in time; $x$ is the horizontal coordinate; $t$ is the time; $k$ is the thermal conductivity of ice dependent on $T$; $\rho$ is the density of ice; $c$ is the specific heat capacity of ice dependent
on $T$; $w$ and $u$ are, respectively, the vertical velocity and horizontal velocity of the ice sheet dependent on $z$ and $t$.

The temperature measured at the six drill sites can be considered at thermal steady state in their near base portion. Three drill sites (Dome C, Dome F, Vostok) are in close vicinity to ice divides where horizontal advection and horizontal heat conduction are assumed to be minimal and the environment approximates a steady state (Cuffey and Paterson, 2010). To first order, we also assume WAIS Divide is in a steady state. Byrd and Kohnen are in the interior slow-moving areas of the Antarctic Ice Sheet with a relatively smooth bed, where horizontal conduction is much lower than vertical conduction (Hindmarsh, 1999, 2018), and horizontal advection and horizontal heat conduction can be neglected (Robin, 1955; Van Liefferinge et al., 2018). This assumption reduces the non-steady-state heat-transfer equation to

$$\frac{\partial}{\partial z}\left(k\frac{\partial T}{\partial z}\right) - \rho c w \frac{\partial T}{\partial z} = 0, \tag{2}$$

which can be rewritten as

$$\frac{1}{k}\frac{\partial k}{\partial z}\frac{\partial T}{\partial z} + \frac{\partial^2 T}{\partial z^2} - \frac{\rho c}{k}w\frac{\partial T}{\partial z} = 0. \tag{3}$$

Using $\frac{\partial k}{\partial z} = \frac{\partial k}{\partial T}\frac{\partial T}{\partial z}$, Eq. (3) becomes

$$\frac{\partial^2 T}{\partial z^2} + \left(\frac{1}{k}\frac{\partial k}{\partial T}\frac{\partial T}{\partial z} - \frac{\rho c}{k}w\right)\frac{\partial T}{\partial z} = 0. \tag{4}$$

According to Fischer et al. (2013), in the most general terms, the vertical velocity in the ice can be approximated by

$$w(z) = -w_{melt} - (Acc - w_{melt})\left(\frac{z}{H}\right)^{m+1}, \tag{5}$$

where $w_{melt}$ is the basal melt rate; $Acc$ is the surface accumulation rate dependent on $t$; and $m$ is an adjustable form factor that accounts for the variation of horizontal velocity.

Substitution of Eq. (5) into Eq. (4) and integrating on the assumption that $k$ is constant gives the following temperature distribution in ice sheet at steady state:

$$T = T_s - \left[\frac{\partial T}{\partial z}\right]_B \int_0^z exp\left(\frac{(w_{melt}-Acc)z^{m+2}}{\alpha_T(m+2)H^{m+1}} - \frac{w_{melt}}{\alpha_T}z\right)dz + \left[\frac{\partial T}{\partial z}\right]_B \int_0^H exp\left(\frac{(w_{melt}-Acc)z^{m+2}}{\alpha_T(m+2)H^{m+1}} - \frac{w_{melt}}{\alpha_T}z\right)dz, \tag{6}$$

where $T_s$ is the surface temperature; $\left[\frac{\partial T}{\partial z}\right]_B$ is the temperature gradient at the ice sheet base; $\alpha_T = k/\rho c$ is the thermal diffusivity of ice.

The least squares method was used to fit measured borehole temperatures with this equation. In fitting, the initial values of the unknown parameters $T_s$, $\left[\frac{\partial T}{\partial z}\right]_B$, $Acc$ and $w_{melt}$ can only be guessed and this results in unavoidable uncertainty of fitting. To overcome this large uncertainty, a common genetic algorithm was used to find the optimal global solution of temperature fitting by constraining these unknown parameters to a predetermined range (Reeves and Rowe, 2002). GA can solve optimization problems by limiting the unknown parameters to a predetermined range with any type of constraints, including integer constraints. In general, GA generates high-quality solutions for optimization problems and search problems.

In the GA, the crossover fraction is set to be 0.9 while the migration fraction is 0.2 (Reeves and Rowe, 2002). To obtain an accurate solution and save calculation time, we set the population size to be 8000 and the number of generations to be 20. Usually, after 15 generations of iteration, the optimal solution can be found. All the calculations were performed using

MATLAB software. GA provides results for the first generation of optimal solution in a wide range based on a random combination of the fitting parameters. Thus, for each deep borehole, the fitting experiments were trialed five times to avoid random error of the GA caused by the initial random parameter combination. Then, the average value from the five fitting experiments was used as the GHF from bedrock into the ice sheet at the selected site.

Equation (4) can also be re-expressed as follows:

$$w(z) = \left[\alpha_T(\frac{\partial^2 T}{\partial z^2}) - \alpha_T(\frac{1}{k}\frac{\partial k}{\partial T})\left(\frac{\partial T}{\partial z}\right)^2\right]/\frac{\partial T}{\partial z}. \tag{7}$$

Note that the vertical velocity is markedly affected by $\frac{\partial^2 T}{\partial z^2}(z)$ and $\frac{\partial T}{\partial z}(z)$. At the base of the ice sheet, the melt/freezing rate is $w_{melt} = w(0)$ while the gradient is $\left[\frac{\partial T}{\partial z}\right]_B = \frac{\partial T}{\partial z}(0)$. The geothermal heat flux $Q_{geo}$ from below the ice is balanced by the conductive flux in the ice $q = k\left[\frac{\partial T}{\partial z}\right]_B$ and the rate at which energy is used to melt/freeze ice, $J = \rho L w_{melt}$. Thus, the GHF will

be:

$$Q_{geo} = \rho L w_{melt} - k\left[\frac{\partial T}{\partial z}\right]_B, \tag{8}$$

where $L$ is the specific latent heat for melting of ice.

**2.3 Uncertainties**

In our method, the temperature in the lower portion of the ice sheets is assumed to be in steady state and the GA algorithm is
used to fit the measured temperatures in deep ice-core drilling boreholes by varying the four key parameters influencing the temperature distribution: the surface temperature, surface accumulation rate, basal melt, and basal temperature gradient. All these parameters are suggested by algorithm in order to obtain the best-fitting curve. We assume that the main uncertainties in our fitting model come from temperature measurements, variability of the form factor $m$, ice thickness estimation and GA algorithm itself. It must be noted that the uncertainties we state are lower limits. There are some additional unexamined
uncertainties that were missing from our model including transient effects associated with climate change and ice-sheet dynamics, the horizontal velocity field, the form of the vertical velocity field, the temperature dependence of the thermal conductivity, unaccounted for thermal disturbances due to drilling processes, 2D effects, and some other phenomena.

**2.3.1 Temperature measurements**

Interpretation of temperature measurements in mechanically drilled deep boreholes filled with drilling fluids is complicated
by several factors (Clow, 2008). First, the temperature is measured in the borehole fluid, not in the surrounding ice; therefore, an important consideration is the need for thermal equilibration of the ice wall and the borehole fluids following drilling and prior to measurement. Second, the heat produced during drilling needs to be dissipated from the borehole or the thermal drilling disturbance needs to be accounted for (Clow, 2015). Third, increasing temperature with depth can cause convective mixing in the borehole. Fourth, the depth of temperature measurements has an inherent uncertainty due to cable slippage in the counting

assembly and cable elongation. Thus, all successful temperature measurements in deep boreholes obey a logging protocol in terms of logger tripping speed, measurement direction, borehole settling time, and so on to minimize the effects of these complicating factors. Temperature measurement errors from sensor accuracy and calibration are found to be within the tolerance for large-scale GHF estimates for our six boreholes to interpret ice-sheet basal dynamics.

The temperature in the Byrd borehole was measured with an accuracy of 0.1 °C (H. Ueda, personal communication).
Motoyama et al. (2013) reported that temperature measurements at Dome F were carried out with a precision of 0.05 °C. The absolute temperature measurement error at Vostok was estimated to be 0.07 °C (Salamatin et al., 1998a). The resolution of the temperature measurements in the Dome C borehole was 0.015 °C, while the precision was found to be 0.05 °C (Leferbre et al., 2002). The logger used in the borehole at Kohnen was calibrated to 0.03 °C (Gundestrup et al., 1994). Prior to drilling, a detailed study of the expected temperature-measurement uncertainties was made for the WAIS Divide site to optimize the
logging system setup (Clow, 2008).  The standard uncertainty (accuracy) of the subsequent WAIS Divide temperature logs was ~0.0053 °C (Cuffey et al., 2016). In general, temperature measurement accuracy in the studied boreholes is more than adequate, and the measured drilling depth was recalculated to true vertical depth using available borehole inclinations.

### 2.3.2 Form factor $m$

Selection of the appropriate form factor $m$ is a challenging task. Classically, vertical velocity depends linearly on $z/H$ (Cuffey
and Paterson, 2010) and $m = 0$. However, at an ice divide, the downward flow of ice is slower, for the same depth, than at locations away from the divide (Raymond, 1983). This reduces the cooling influence of vertical advection and increases the basal temperature. Therefore, Raymond (1983) suggested the use of $m = 1.0$ for deformation in the vicinity of ice divides.

To set up the vertical velocity profile at Dome C, Fischer et al. (2013) performed three runs with $m = 0.3$, $m = 0.5$ and $m = 0.7$ and found that the temperature profile is only slightly affected by this choice. However, the form factor $m$ exhibited a strong
influence on the age profile of the ice. That was the reason why the authors used $m = 0.5$, which is in good agreement with the EDC3 age scale (Parrenin et al., 2007b). Following the method of Fischer et al., (2013), we modeled the age profile of ice by

$$\hat{A} = \int_{z}^{H} \frac{1}{w_{melt} + (Acc - w_{melt})\left(\frac{z}{H}\right)^{m+1}} dz, \tag{9}$$

where $\hat{A}$ is the modeled ice age. Then, the modeled ice age is used to compare with the published depth-age scales at the studied sites and the best value for the form factor $m$ is estimated. In order to reduce the run time of multilevel calculations,
we examine the form factor $m$ at only five levels 0, 0.25, 0.50, 0.75, and 1.00. The best value for the form factor $m$ is selected on the basis of the nonlinear correlation analysis between modeled and measured age scales. To calculate the correlation factor $R^2$, we first found the average value of the measured age:

$$\bar{A} = \frac{1}{n}\sum_{i=1}^{n} A_i, \tag{10}$$

where $n$ is the number of measured ice ages $A_i$. Then the total sum of squares $SS_{tot}$ and the sum of squares of residuals $SS_{res}$
were calculated:

$$SS_{tot} = \sum_{i=1}^{n}(A_i - \bar{A})^2, \tag{11}$$

$$SS_{res} = \sum_{i=1}^{n}(A_i - \hat{A}_i)^2, \tag{12}$$

where $\hat{A}_i$ is the modeled ice age when $n = i$. The correlation factor $R^2$ was estimated by

$$R^2 = 1 - \frac{SS_{res}}{SS_{tot}}. \tag{13}$$

Finally, the results of the nonlinear correlation analysis were checked by evaluating the root mean squared error (RMSE):

$$\text{RMSE} = \sqrt{\frac{1}{n}\sum_{i=1}^{n}(A_i - \hat{A}_i)^2}. \tag{14}$$

### 2.3.3 Ice thickness

We assume that the ice sheet thickness at the studied sites has kept constant at the present day height; however, it has varied in the past. The 3D thermo-mechanical model and the simple 1D model showed that the maximum variation of ice sheet
thickness at Dome C and Dome F was less than 250 m in the past (Parrenin et al., 2007a). In general, the typical difference in the ice thickness in the glacial and interglacial periods at Dome C was 150 m (Passalacqua et al., 2017). At the Kohnen site, the local elevation variation is on the order of 100 m (Huybrechts et al., 2007). The ice thickness variation at Vostok, located in the central part of East Antarctica plateau, exhibits the similar range as at Dome F and Dome C (Ritz et al., 2001).

The best evidence for ice-sheet elevation change in the interior of the West Antarctic ice sheet comes from the Ohio Range, to
the south of the WAIS Divide site at a height of 1600 m a.s.l., and from Mt. Waesche to the north of the WAIS Divide site at a height of 2000 m a.s.l. (Ackert et al., 1999, 2007). Moraines at Mt. Waesche were ~50 m higher and trimlines in the Ohio Range were ~125 m higher, between 12 and 10 ka. The thinning of ~100 m throughout the Holocene occurred as the grounding line retreated by hundreds of km and the accumulation rates were relatively stable (Anderson et al., 2002; Conway et al., 1999). Cuffey et al. (2016) presented a model which indicates a more likely scenario of 200 m thickening at WAIS Divide when the
accumulation rate rose after the last glacial maximum, followed by 300 m of thinning to the mid-Holocene. The elevation change is comparable to the amount of elevation change inferred for interior East Antarctic sites.

Comparison with the modern ice thickness value indicates that the variation of ice thickness is small and its influence on ice temperature distribution can be neglected, in particular, on lower portion of the ice borehole. For example, assuming a 150 m thickness increase from the LGM to 15 ka leads to the change in the reconstructed LGM temperature by less than 0.2 °C
compared to a constant thickness in WAIS ice core (Buizert et al., 2015). This is the reason why constant ice thickness is also used by other researchers for GHF estimates (Dahl-Jensen et al., 2003; Engelhardt, 2004; Mony et al., 2020).

### 2.3.4 Genetic algorithm

For each deep borehole, the fitting experiments were repeated five times for the best value of the form factor m and the average value obtained from the five fitting experiments was used as the representative value of GHF from bedrock into ice. Thus, the
uncertainty ranges came from the difference between the maximum/minimum and the average GHF values.

## 3 Results

### 3.1 Initial conditions and GHF estimates

GHF estimates were made using the following ice parameters:

- density 918 kg m$^{-3}$;
- specific heat capacity $c = 152.5 + 7.122 (T + 273.15)$ J kg$^{-1}$ K$^{-1}$ (Yen, 1981);
- thermal conductivity $k = 9.828e^{-0.0057(T+273.15)}$ W m$^{-1}$ K$^{-1}$ (Yen, 1981);
- specific latent heat L = 333.5 kJ kg$^{-1}$ (Cuffey and Paterson, 2010).

In our model, we assume that $k$ and $c$ are constant and equal to their values at the temperature of the pressure melting point; this can provide a better estimate of the basal melting rate at the base of ice sheet (Fischer et al., 2013). In this case, $\frac{1}{k}\frac{\partial k}{\partial T}$ =-5.7e-3 K$^{-1}$. Figure 3 shows the fitted temperature profiles compared with measured temperatures.

We performed five runs for estimating GHF with $m = 0$, $m = 0.25$, $m = 0.5$, $m = 0.75$, and $m = 1.0$ for each site and compared modeled and measured age scales. As illustrated in Fig. 4, the form factor has a strong influence on the age profile of the ice. The results of our estimates for GHF for different $m$ values are summarized in Table 4. Surprisingly, on three separate occasions the correlation factor is negative. This may occur when $SS_{res}$ is far beyond that of $SS_{tot}$ (see Eqs. (10)-(12)). That is to say that there is no correlation between modeled and measured age scales in these cases. The results of the correlation analyses were confirmed by evaluating how close the modeled lines are to the data points with the aid of the RMSE. The smaller the RMSE, the closer the model is to the data. The GHF and $m$ values with the highest correlation factor and smallest RSME were selected for further processing and trialed five times. The average GHF values for selected $m$ are added into Table 2. The precision of the GHF estimates and basal melt/freezing rate are also specified here.

The temperature profiles show that the heat flow through the ice at six deep drilling sites in Antarctica must be >42.6–77.1 mW m$^{-2}$ in order to match the observed temperatures in the boreholes. The basal ice at all sites is at the pressure-melting point, and the amount of melt cannot be constrained by the energy-balance equation alone. When the heat flow model is combined with vertical velocity estimates, the estimated heat flow can be translated to a GHF of 57.9–113.3 mW m$^{-2}$, except for Vostok with GHF of -3.3 mW m$^{-2}$.

### 3.2 Data comparison and divergences

*Vostok*. The surface temperature-time curve for the upper bound of the present-day accumulation rate at Vostok corresponds to a GHF of 53 mW m$^{-2}$ (Salamatin et al., 1998b). We calculated that at the base of the ice sheet, the conductive flux is 42.6±0.4 mW m$^{-2}$ while the latent heat flux from refreezing of the lake water is 46.3±5.6 mW m$^{-2}$. Thus, the GHF heat flux at the base of the ice sheet has a negative value of -3.6±5.3 mW m$^{-2}$. This is in good agreement with the isotope studies that showed that the Vostok ice core consists of ice refrozen from Lake Vostok water, from 3539 m below the surface of the Antarctic ice sheet to its bottom (Jouzel et al., 1999). Sufficiently high correlation factor (0.75) between modeled and measured age scales at $m =$

1 indicates that ice above Lake Vostok reasonably fits Raymond's (1983) arguments for deformation in the vicinity of ice divides.

At this stage we are not yet able to predict GHF at the bed of 600-m thick subglacial Lake Vostok because the temperature profile in the lake is still indefinite. However, the DNA detection of thermophile bacterium in the near-base accretion ice suggests the existence of near-bottom warm waters with temperatures as high as 50 °C (Bulat et al., 2012). If so, the GHF in the lake sediments can reach 200-240 mW m$^{-2}$. These values can be considered as paleo-GHF because microorganisms were picked up thousands of years ago but still actual accounting for long duration of geological processes.

*Dome C*. The inverse approach to retrieving GHF from radar inferred distribution of wet and dry beds at the EPICA drilling site (Passalacqua et al., 2017) gave 54.5±3.5 mW m$^{-2}$, slightly lower than estimates derived from borehole temperature profiling (57.9±6.4 mW m$^{-2}$). The modeled GHF range (43–55 mW m$^{-2}$ obtained by An et al., 2015; Fox Maule et al., 2005; Shapiro and Ritzwoller, 2004; Van Liefferinge and Pattyn, 2013) is also a little less than our estimates. The high value for the correlation factor (0.997) indicates a perfectly strong relationship between modeled and measured depth-ages scales meaning that there is no horizontal advection of heat and the drill site is located at a perfect dome position. Perhaps, that is the reason that the core from Dome C contains the oldest continuous climate record obtained from ice cores so far (Parrenin et al., 2007b). However, a high spatial variation of GHF at Dome C area was found from radar-sounding data (Carter et al., 2009). The values of nearly 100 mW m$^{-2}$ inferred for the southern shore of Concordia Subglacial Lake, approximately 50 km to the south of the drilling site, are also well outside modeled estimates.

*Kohnen*. The model with a standard GHF of 54.6 mW m$^{-2}$ predicted a basal temperature 0.3 °C below the pressure melting point at Kohnen (Huybrechtset al., 2007). GHF obtained by non-thermal geophysical models are in the range of 46-62 W m$^{-2}$. Our estimate (86.9±16.6 mW m$^{-2}$) is higher than the modeled GHF values suggested by Fox Maule et al. (2005), Martos et al. (2017) and other referenced models. However, subglacial water entering the borehole indicated that the actual GHF should be much higher than that indicated by the regional models. Under these circumstances, our estimate is likely to be closer to the real heat flux. Surprisingly, the depth-age scale is only slightly affected by the choice of the form factor indicating that the variation of horizontal velocity is low at this site.

*Dome F*. A previously estimated GHF of 59 mW m$^{-2}$ neglected the bottom ice melt rate (Hondoh et al., 2002) and thus is lower than our estimate (78.9±5.0 mW m$^{-2}$). Mony et al. (2020) estimated the GHF in Dome F borehole to be even lower at 50.4 mW m$^{-2}$. As the drill approached the base (approx. 10 m above), subglacial meltwater leaked into the borehole and froze onto the drill, directly indicating that ice reaches the pressure melting point, placing a lower bound on the GHF. GHF values obtained by non-thermal geophysical methods are in the range of 48-65 W m$^{-2}$, also lower than our estimates. The correlation factor between modeled and measured depth-age scales is quite high (0.83) at $m$ = 1 indicating ice at the site can be adequately approximated by the steady-state model. Thus, the slightly elevated heat flow at this location appears to represent a regional value.

***Byrd***. Unfortunately, age scales for the Byrd borehole for all modeled *m* values are quite far from the measured depth-age
data. Tilting measurements (Garfield and Ueda, 1976; Hansen et al., 1989) and modeling (Whillans, 1979) showed that the
relative horizontal velocity of ice at this borehole reaches $\sim$3 m a$^{-1}$ at the 1500 m depth. Thus, horizontal conduction in the
bottom of the ice sheet is quite high at this site, producing a high divergence from the steady-state model. Even the correlation
factor for the best fitting age-depth curve with *m* = 0.75 is only 0.58, we can use the GHF value of 88.4±7.6 mW m$^{-2}$ at this
location as a reference until more precise estimates are obtained. This value is higher than the first estimate made immediately
after temperature logging (75.4 mW m$^{-2}$ referenced by Ueda, 2007), primarily because the latter one did not account for the
basal ice melt. The latest modeling by Martos et al. (2017) revealed a high GHF at the location of Byrd Station (132 mW m$^{-2}$
with an error of ±5 mW m$^{-2}$) when compared with values obtained from previous models (An et al., 2015; Fox Maule et al.,
2005; Van Liefferinge and Pattyn, 2013). Generally, our approximate estimate is in the range of GHF values suggested by
previous models.

***WAIS Divide***. A preliminary estimate of the GHF at this site suggested a high value in the range of 200-230 mW m$^{-2}$, depending
on the actual ice thickness (Clow et al., 2012). This is more than twice that derived using non-thermal geophysical methods.
There is no depression in the local surface topography or drawdown in the subsurface layers detected by ice-penetrating radar,
as would be expected over a local hot spot. Using airborne magnetic data, Martos et al. (2017) estimated the highest value for
this area to be $\sim$120 mW m$^{-2}$. Our new estimate is slightly lower at 113.3±16.9 mW m$^{-2}$. Mony et al. (2020) also estimated
GHF from the borehole temperature profile at WAIS Divide by combining a heat-transfer equation and the physical properties
of the ice sheet in a numerical model. Based on a truncated temperarture profile, they estimate a GHF of 90.5 mW m$^{-2}$ which
is less than ours and fairly corresponds to the latest GHF map for Antarctica constructed by empirically relating the upper
mantle seismic structure (Shen et al., 2020). The low correlation factor (0.59) between modeled and measured depth-age scales
in our present estimate indicates that there are some important uncertainties that the steady-state model does not account for.
Most likely, these are the same unaccounted effects that affect the Byrd borehole temperature profile, i.e., horizontal flow and
climate-related transient effects.

The preliminary GHF estimate (Clow et al., 2012) was based on the first temperature log in 2011 in the borehole before it
reached its final depth. The reasons why the preliminary GHF estimate may be so high are that: (i) temperatures in the borehole
were still thermally disturbed in 2011, and (ii) the bottom of the 2011 temperature log was still far from the base of the ice
sheet. The borehole was relogged in 2014, and temperature data were obtained much closer to the bed. In addition, Clow et al.
(2012) also did not account for horizontal flow effects and the GHF estimation could have been lower than the one they
produced. Further investigations on ice dynamics through WAIS Divide borehole tilt measurements can allow us to determine
in-depth stress and velocity distributions and estimate horizontal flow effects on temperature.

### 3.3 Indirect results

Although a steady-state model is used in the lower portion of the boreholes to describe the temperature distribution, it is worth noting that the measured modern temperature is the cumulative effect of historical climate forcing. Therefore, the best fitting parameters obtained by GA are not the real parameters occurring during the ice sheet's history. They can be considered as "equivalent" parameters which are used for calculating the modern temperature profile by eliminating the historical climate changes. Processing back, the "equivalent" vertical velocity, modern accumulation rate and temperature can be calculated from

the GA results. Estimated vertical velocity profiles are shown on Fig. 5. Table 5 lists values of "equivalent" snow accumulation rate and temperature at ice sheet surface which were derived from GA calculations. In all cases, "equivalent" accumulation rates are higher than the modern rates while the "equivalent" surface temperatures are very close to the modern ones. This can be explained by the fact that the high "equivalent" accumulation rates are used by GA to eliminate the colder climate effects on the ice temperature profile during the glacial period.

## 4 Discussion

### 4.1 Transient model vs. steady-state model

Both, transient thermal models (e.g., Dahl-Jensen et al., 2003; Engelhardt, 2004; Martos et al., 2017; Passalacqua et al., 2017; Van Liefferinge et al., 2018) and steady-state models (e.g., Martin and Gudmundsson, 2012; Mony et al., 2020; Parrenin et al., 2017; Price et al., 2002; Zagorodnov et al., 2012) were used intensively in the past and are still used for GHF estimates in

Antarctica. Obviously, an exact steady state never occurs in reality and thus transient models would be expected to give more precise results than steady-state models. However, the answer is not as simple as it is supposed to be.

It is important to recognize that, first, in both cases the models will produce GHF "estimates", not "measurements", and second, the thermal gradient can be affected by processes other than GHF, creating local anomalies that may coincide with the point estimate. In order to use a transient model, the accumulation rate and surface temperature in the past should be known. For

some of the discussed drill sites these data are available from ice-core studies, while for other sites they are not.

To evaluate the possibility of using a transient model, the GHF at WAIS Divide was estimated by using the accumulation rate and surface temperature in the past provided by Buizert et al. (2015). In these calculations, we assume that the history of the ice sheet at WAIS Divide is about 68 ka long. The governing equation for the transient model was solved using the finite difference method. The equation was discretized by both the central-difference and upwind-difference methods and then solved

using Matlab. To find the best solution, the GA algorithm was still used. The central-difference method and upwind-difference method demonstrated the same temperature profile. Therefore, here we present the calculation results obtained via the upwind difference method.

Unfortunately, the calculation results with the transient model showed the best-fit GHF value of ~500 mW m$^{-2}$ when $m = 1$, which seems to be unrealistic. Moreover, after running the model, we found that after about 4-8 ka, the influence of initial

temperature on temperature profile can be ignored. Later, we assumed the form factor $m = 0$ and a GHF value of 235 mW m$^{-2}$ which showed a good fit with the measured temperatures, although the GHF is much higher than our earlier estimate and estimates from regional models. The temperature distribution in history was modeled 61.2 ka, 54.4 ka, … 6.8 ka ago until modern day (Fig. 6). As expected, the modeled temperature in the upper part of the ice sheet grossly changes with time but in the lower portion (~1000 m above ice sheet base) these variations are much smaller. This indicates that the heat disturbance

related to atmosphere forcing (temperature and snow precipitation) from the ice-sheet surface gradually decays with the depth. From all appearances, the near-basal portion is close to a steady state.

Both models lack additional heat sources (i.e., shear heating, heat advection and basal frictional heating) that might be generated at the bottom of the ice sheet. Thus, the results of both modeling approaches strongly depend on the selected initial parameters, in particular, from the selected value for the form factor $m$. Experimental validation of both models for adequacy

is extremely difficult and recently both of them have rights to exist if assumptions are examined analytically.

### 4.2 Implications of elevated heat flux

Most numerical models of the EAIS basal conditions assume the GHF to be 42-65 mW m$^{-2}$. However, the presence of basal meltwater beneath most of the Antarctic ice sheet requires GHF $\geq 80$ mW m$^{-2}$ (Budd et al., 1984). In support of this conjecture, processing of available temperature profiles in ice boreholes shows that at sites where subglacial water exists bedrocks are

quite warm. Currently, a warm base beneath the EAIS was also confirmed by tectonic reconstructions (Carson et al., 2014). Additional evidence of high GHF at EAIS locations comes from ice-penetrating radar data that revealed a ~100 km long and 50 km wide area near South Pole with GHF of 120±20 mW m$^{-2}$, more than double the values expected for this cratonic sector of East Antarctica (Jordan et al., 2018). This warm base could be caused by radiogenic heat effects or hydrothermal circulation, but a coherent explanation for this phenomenon is still required.

Variability of crustal thickness, hydrothermal circulation (Seroussi et al., 2017), magmatic intrusion (Van Wyk De Vries et al., 2017), and thermal conductivity variability are the main contributors to the elevated and highly variable values of GHF in West Antarctica (Begeman et al., 2017). One of the first pieces of evidence for an "unreasonably high" GHF (>100 mW m$^{-2}$) under WAIS came from temperature-depth profiles in a 480-m-deep borehole drilled at Crary Ice Rise, Antarctica (Bindschadler et al., 1990). Measured GHF in Subglacial Lake Whillans was found to be $285 \pm 80$ mW m$^{-2}$, significantly

higher than the continental and regional averages estimated for this site by using geophysical and glaciological models (Fisher et al., 2015). The GHF at the vents of subglacial volcanoes in West Antarctica can be as high as 25 W m$^{-2}$ and one such 23-km-wide caldera was revealed ~100 km to the south of the WAIS Divide drill site (Blankenship et al., 1993). Undeniably, more systematic explorations are still required to study how far this heat flow high extends into the interior of the West Antarctic Rift System.

## 5 Conclusions

Prediction of the future behavior of the Antarctic Ice Sheet undeniably requires accurate ice-sheet models. However, GHF models based on seismic tomography, radar data, magnetic field observations, the tectonic age and geological structure of the bedrock yields mixed results at sites of deep ice-core drilling in Antarctica. We suggested to estimate GHF from ice-borehole temperature profiles using a one-dimensional steady-state energy-balance equation and the genetic algorithm (GA) for determining the optimal solution of temperature fitting. To our knowledge, we used the GA approach for the first time in ice thermodynamics. Comparison of modeled and measured depth-age scales show that our model is able to assess the variation in GHF estimates from ice-borehole temperature profiles if in-depth horizontal ice velocities are low and can be ignored. The correlation analyses at the EAIS sites indicates that all of them can be adequately approximated by the steady-state model. However, horizontal velocities and their variation over ice-age cycles are much greater at WAIS than at the EAIS sites. Thus, the steady-state model cannot precisely describe temperature distribution here.

At three studied EAIS sites (Dome C, Dome F, and Kohnen), the GHF is higher than that predicted by other models. We assume that this elevated GHF can represent regional value and can be used as a reference point for regional modelings. More precise GHF estimates and explanations for an elevated GHF would be possible after temperature logging and subglacial rock studies from deep boreholes that are required to drill in Antarctica in the distant future. Finally, the proposed method of GHF estimates can be used at other sites in Antarctica and Greenland where the steady-state model is acceptable.

## Author contribution

PT and YL developed the concept, wrote the manuscript and drew all figures. YL also designed and performed all GHF estimations. JH processed all temperature profiles. GDC has made important recommendations and edited the final version of the manuscript. LA, GDC, EL, AM, HM, and CR provided observed borehole temperature data. All authors contributed to discussion and interpretation of the results.

## Competing interests

The authors declare that they have no conflict of interest.

## Acknowledgments

This work was supported by grant Nos. 41327804 and 41806220 from the National Natural Science Foundation of China and the Program for Jilin University Science and Technology Innovative Research Team (Fundamental Research Funds for the Central Universities, Project No. 2017TD-24). We are grateful to H. Ueda (retired from USA CRREL) for providing original Byrd borehole temperature-log data. We thank D. Dahl-Jensen (Centre for Ice and Climate, University of Copenhagen,

Denmark) for fruitful discussion and useful comments. We also would like to thank the Editor Alex Robinson and both anonymous reviewers for fruitful comments and advices.

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

**Table 1: Information for Antarctic deep ice-drilling sites**

| Parameters | WAIS | | EAIS | | | |
|---|---|---|---|---|---|---|
| | **Byrd** | **WAIS Divide** | **Vostok** | **Dome C** | **Kohnen** | **Dome F** |
| Coordinates | 80°01′ S, 119°31′ W | 79°28′ S, 112°05′ W | 78°28′ S, 106°48′ E | 75°06′ S, 123°24′ E | 75° S, 0° E | 77°19′ S, 39°40′ E |
| Years drilled | 1966-1968[a] | 2006-2011[d] | 1990–1998, 2005–2014[f,g] | 1999-2004[i] | 2002-2006[k] | 2003-2007[n] |
| Surface elevation (m a.s.l.) | 1530[a] | 1766[e] | 3488[f] | 3233[j] | 2892[l] | 3810[j] |
| Drilled depth (m) | 2193 | 3405[d] | 3769.3[g] | 3270.2[i] | 2774.2[k] | 3035.2[n] |
| Ice thickness according with radar/seismic survey (m) | 2300[b] | 3455[e] | 3750±20[g] | 3273±5[j] | 2750±50[l] | 3028±15[j] |
| Snow accumulation at surface (mm ice $a^{-1}$) | 169.5[c] | 220[e] | 24.8[h] | 28.4[j] | 70[m] | 29.9[j] |
| Ice sheet surface horizontal velocity, m $a^{-1}$ | 12.7[o] | ~3.0[t,u] | 2.00±0.01[s] | 0.015±0.01[p] | 0.74[r] | Negligible[v] |
| Mean surface snow temperature (°C) | -28[a] | -30[e] | -57[h] | -54.6[j] | -44[l] | -57.3[j] |

[a]Ueda, 2007; [b]Wexler, 1961; [c]Gow, 1968; [d]Slawny et al., 2014; [e]WAIS Divide Project Members, 2013; [f]Vasiliev et al., 2011; [g]Lukin and Vasiliev, 2014; [h]Ekaykin et al., 2012; [i]Augustin et al., 2007; [j]Parrenin et al., 2007a; [k]Wilhelms et al., 2014; [l]Ueltzhöffer et al., 2010; [m]Huybrechts et al., 2007; [n]Motoyama, 2007; [o]Whillans, 1977; [p]Vittuari et al.., 2004; [r]Wesche et al., 2007; [s]Wendt et al., 2006; [t]Conway and Rasmussen, 2009; [u]Koutnik et al., 2016; [v]Motoyama et al., 2008

**Table 2: Thermophysical properties at the base of Antarctic Ice Sheet at sites of deep ice-drilling estimated in this study**

| Parameters | WAIS | | EAIS | | | |
|---|---|---|---|---|---|---|
| | **Byrd** | **WAIS Divide** | **Vostok** | **Dome C** | **Kohnen** | **Dome F** |
| Temperature, °C | -1.43 | -2.30 | -2.49 | -2.15 | -1.85 | -1.99 |
| Temperature gradient (°C 100 m$^{-1}$) | 3.70 | 3.88 | 2.02 | 2.42 | 3.12 | 2.66 |
| Ice thickness according with depth of pressure melting point (m) | 2164 | 3485 | 3759 | 3257 | 2770 | 3016 |
| Basal melt rate (mm a$^{-1}$) | 1.2±0.8 | 3.7±1.7 | -4.8±0.6 | 1.08±0.27 | 2.8±1.6 | 2.5±0.5 |
| GHF (mW m$^{-2}$) | 88.4±7.6 | 113.3±16.9 | -3.6±5.3 | 57.9±6.4 | 86.9±16.6 | 78.9±5.0 |

**Table 3: Polynomial approximations of borehole temperature $T$ (°C) as a function of true vertical depth $z$ and correlation factors**

| Drill sites | Polynomial | $R^2$ |
|---|---|---|
| Byrd | $T = -28.343 + 0.8367 \times 10^{-3} z - 6.7651 \times 10^{-6} z^2 + 6{,}1339 \times 10^{-9} z^3$ | 0.997 |
| WAIS Divide | $T = -31.799 + 8.8595 \times 10^{-3} z - 9.4649 \times 10^{-6} z^2 + 2.657 \times 10^{-9} z^3$ | 0.997 |
| Vostok | $T = -56.034 + 2.9889 \times 10^{-3} z + 3.888 \times 10^{-6} z^2 + 0.2419 \times 10^{-9} z^3$ | 0.999 |
| Dome C | $T = -54.316 + 5.2978 \times 10^{-3} z + 4.4141 \times 10^{-6} z^2 - 0.368 \times 10^{-9} z^3$ | 0.999 |
| Kohnen | $T = -44.428 + 1.7384 \times 10^{-3} z + 4.4124 \times 10^{-6} z^2 + 0.184 \times 10^{-9} z^3$ | 0.999 |
| Dome F | $T = -55.016 + 5.839 \times 10^{-3} z + 5.188 \times 10^{-6} z^2 - 0.446 \times 10^{-9} z^3$ | 0.998 |

**Table 4. GHF (mW m$^{-2}$) calculated for different form factors *m* in the steady-state model, the correlation factor $R^2$ between modeled and measured age scales and RSME**

| Parameters | WAIS | | EAIS | | | |
|---|---|---|---|---|---|---|
| | **Byrd** | **WAIS Divide** | **Vostok** | **Dome C** | **Kohnen** | **Dome F** |
| **GHF for *m* = 0** | 8.72 | 72.9 | -16.16 | 134.6 | 66.04 | 92.78 |
| $R^2$ | -0.37 | -2.46 | 0.65 | 0.69 | 0.978 | 0.78 |
| **RMSE** | 28.19 | 36.40 | 58.95 | 107.51 | 4.132 | 40.62 |
| **GHF for *m* = 0.25** | 28.44 | **100.0** | -35.28 | 105.3 | **60.95** | 130.9 |
| $R^2$ | 0.12 | **0.59** | 0.50 | 0.72 | **0.986** | -1.62 |
| **RMSE** | 22.60 | **12.93** | 70.60 | 102.14 | **2.431** | 140.91 |
| **GHF for *m* = 0.50** | 55.05 | 207.4 | -20.33 | 70.06 | 156.4 | 92.96 |
| $R^2$ | 0.46 | 0.41 | 0.51 | 0.87 | 0.976 | 0.47 |
| **RMSE** | 17.59 | 15.05 | 69.74 | 69.48 | 4.338 | 62.91 |
| **GHF for *m* = 0.75** | **95.84** | 240.3 | -25.39 | **57.40** | 106.0 | 104.8 |
| $R^2$ | **0.57** | 0.32 | 0.39 | **0.997** | 0.984 | 0.53 |
| **RMSE** | **14.86** | 16.15 | 78.01 | **22.34** | 3.477 | 59.25 |
| **GHF for *m* = 1.00** | 117.8 | 251.3 | **-8.90** | 67.3 | 161.5 | **79.20** |
| $R^2$ | 0.45 | 0.29 | **0.75** | 0.95 | 0.982 | **0.83** |
| **RMSE** | 17.83 | 16.45 | **49.71** | 43.31 | 3.692 | **35.87** |

GHF values with the highest correlation factor and smallest RSME are highlighted by bold.

**Table 5: Equivalent thermophysical parameters used by GA in comparison with published data**

| Parameters | Byrd | WAIS Divide | Vostok | Dome C | Kohnen | Dome F |
|---|---|---|---|---|---|---|
| "Equivalent" snow accumulation at surface (cm ice a$^{-1}$) | 52.8 | 48.8 | 8.95 | 3.87 | 6.92 | 3.00 |
| Modern snow accumulation at surface (cm ice a$^{-1}$) | 16.9 | 22.0 | 2.48 | 2.84 | 7.00 | 2.99 |
| "Equivalent" surface temperature (°C) | -29 | -30.9 | -56.5 | -54.7 | -44.9 | -56.5 |
| Modern surface temperature (°C) | -28 | -30 | -57 | -54.6 | -44 | -57.3 |


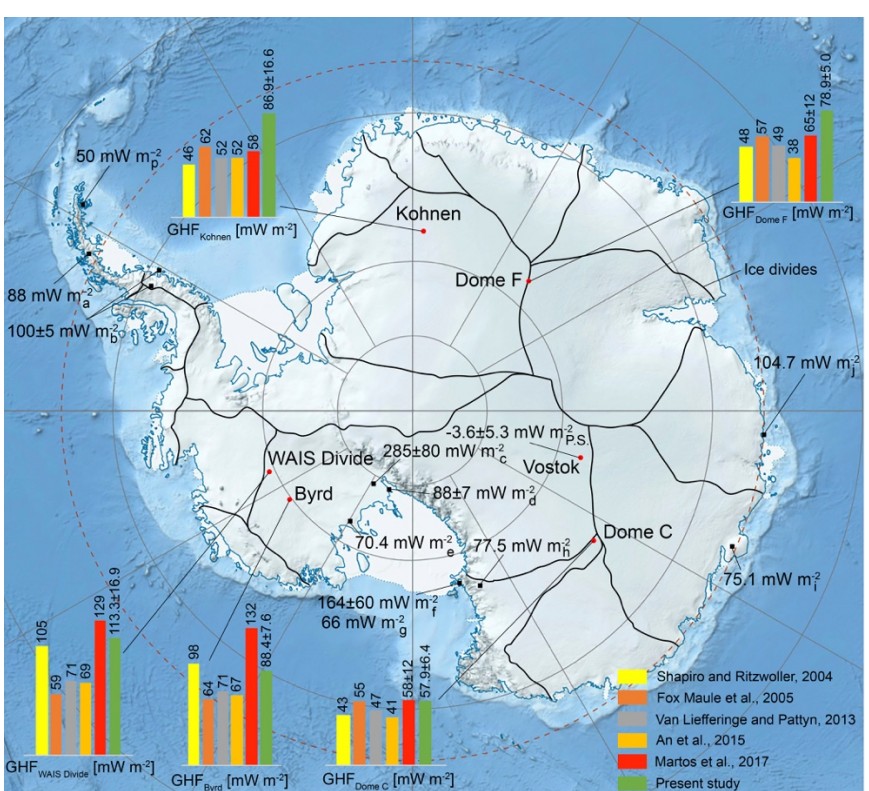

**Figure 1: GHF derived in the present study (P.S.) from basal temperature gradients in deep ice boreholes (green bars) compared with modeling. Red circles show locations of deep ice drilling sites (Byrd, WAIS Divide, Vostok, Dome C, Kohnen, and Dome F) discussed in the present study. Black squares show locations of boreholes drilled in Antarctic margins, in which borehole temperature measurements were carried out and GHF values were estimated ([a]Zagorodnov et al., 2012; [b]Nicholls and Paren, 1993; [c]Fisher et al., 2015; [d]Begeman et al., 2017; [e]Engelhardt, 2004; [f]Risk and Hochstein, 1974; [g]Decker and Bucher, 1982; [h]Clow et al., 2011; [i]Dahl-Jensen et al., 1999; [j]Zotikov, 1961). Drill sites in ocean/sub-shelf sedimentaries are not shown. Location of the Antarctic ice divides is shown according with Rignot et al., 2011.**


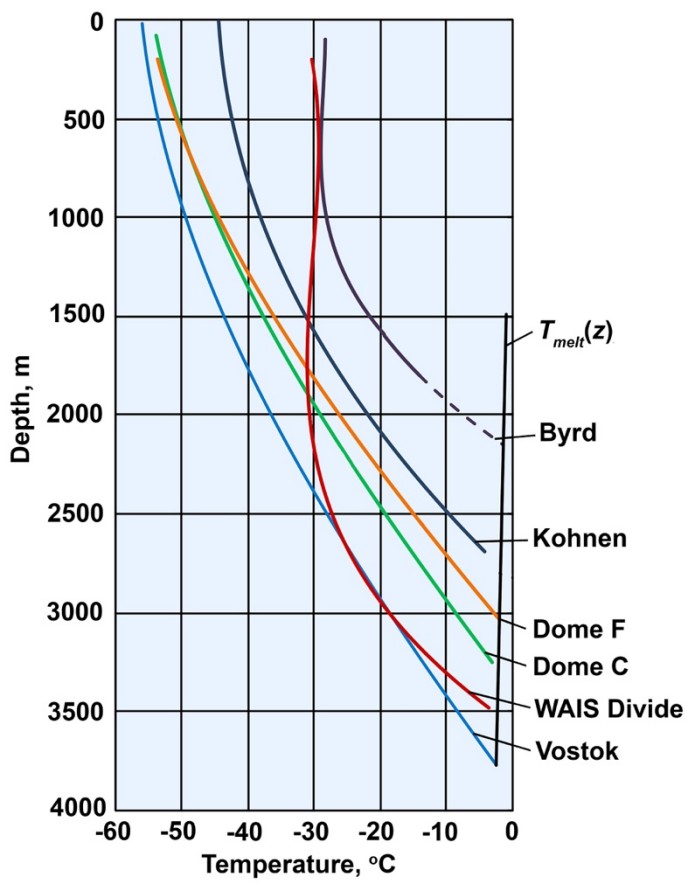


**Figure 2: Smoothed measured temperature profiles in Antarctic deep ice boreholes. Pressure-melting point temperature $T_{melt}(z)$ is shown in the assumption of Clausius-Clapeyron slope of 0.0742 K/MPa.**

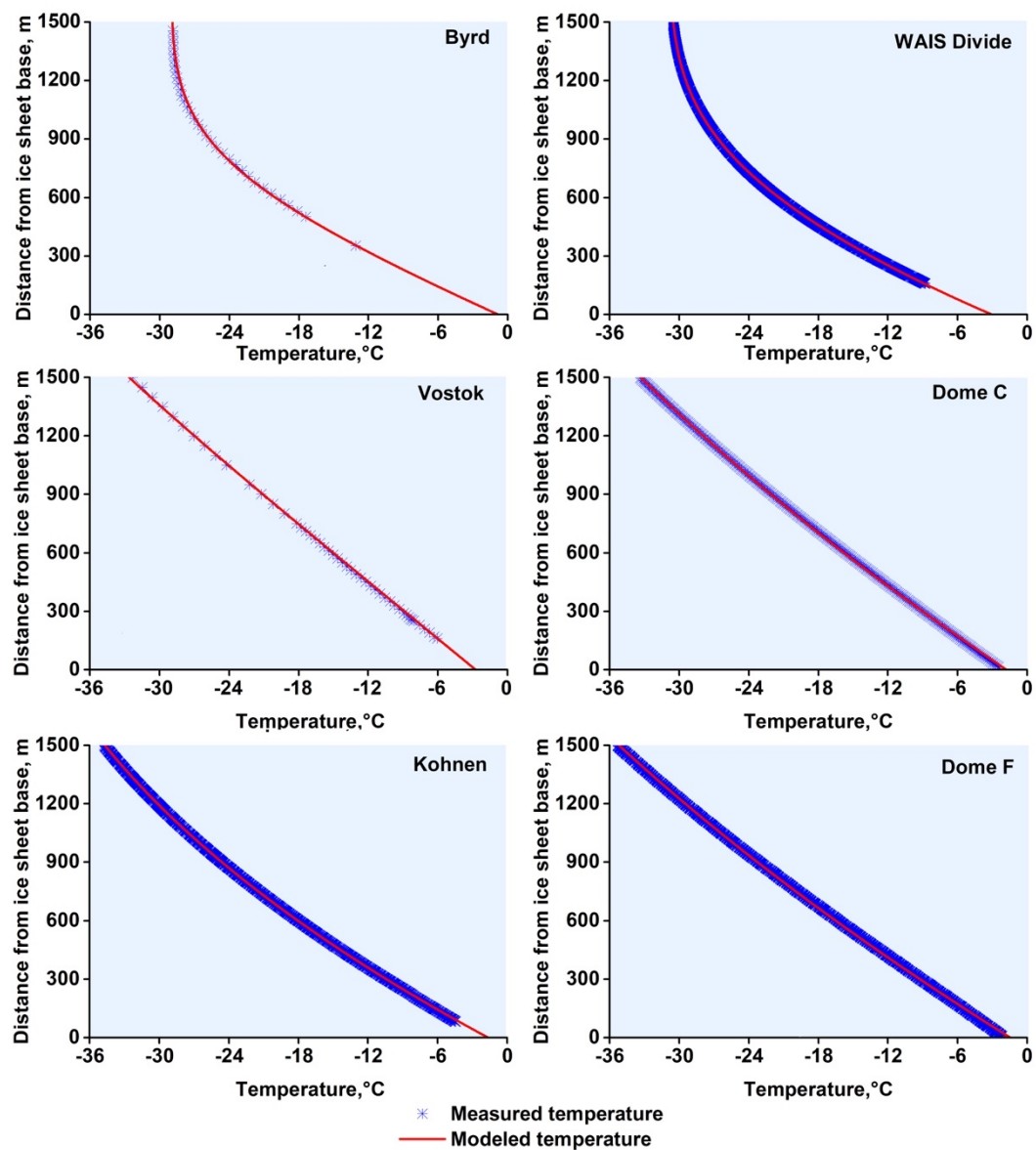

**Figure 3. Temperatures measured in Antarctic deep ice boreholes compared with best-fit temperature profiles for the deepest 1500 m.**

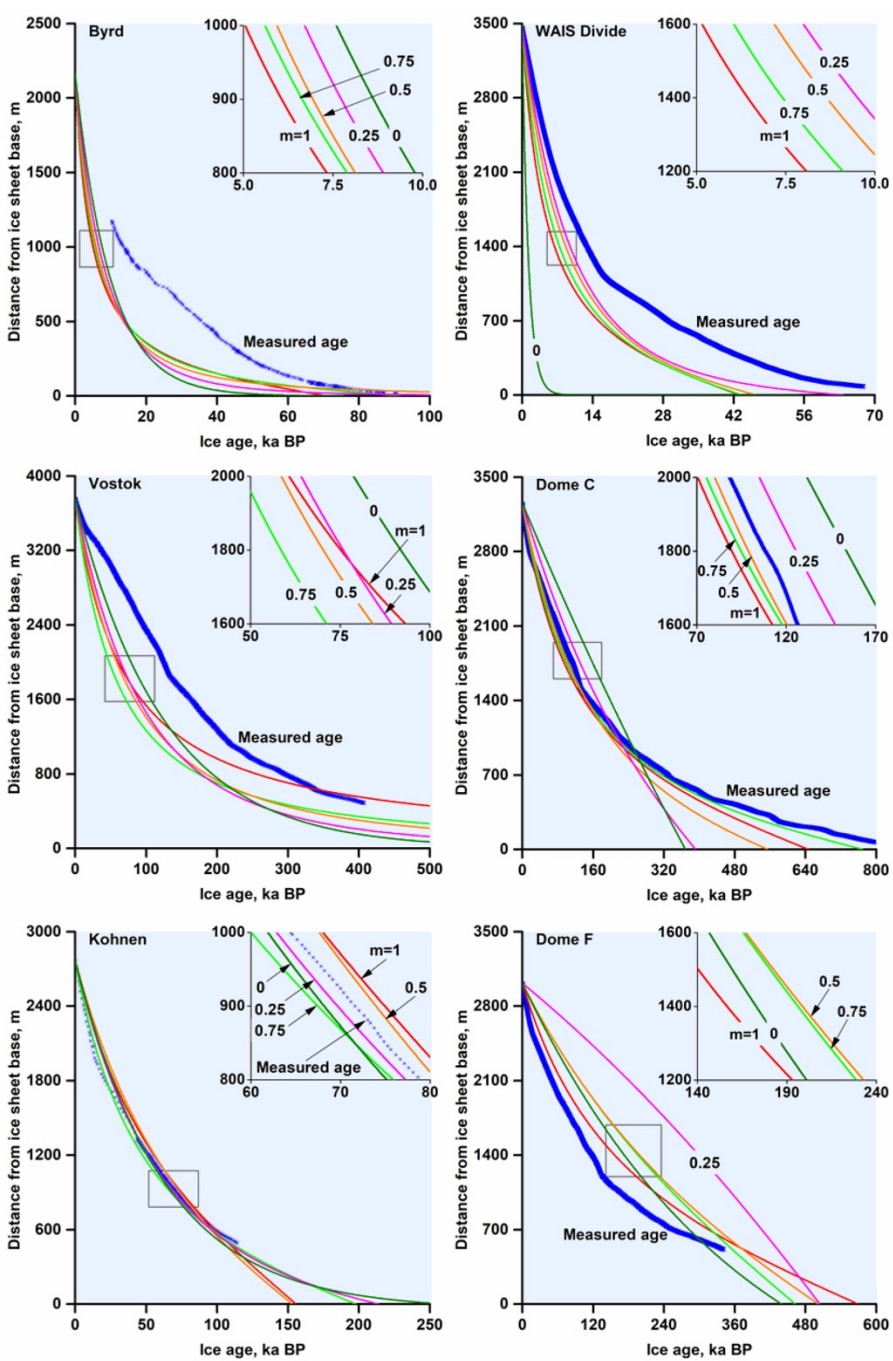


**Figure 4. Comparison of the measured age scales (Ahn and Brook, 2008; Bazin et al., 2013; Bereiter et al., 2012; Blunier and Brook, 2001; Kawamura et al., 2007; Neftel et al., 1988; Parrenin et al., 2007b; Sigl et al., 2016; Staffelbach et al., 1991; Veres et al., 2013) and modeled age scales with $m = 0$, $m = 0.25$, $m = 0.5$, $m = 0.75$, and $m = 1.0$ (correlation factors between modeled and measured age scales for each run are stated in Table 4).**


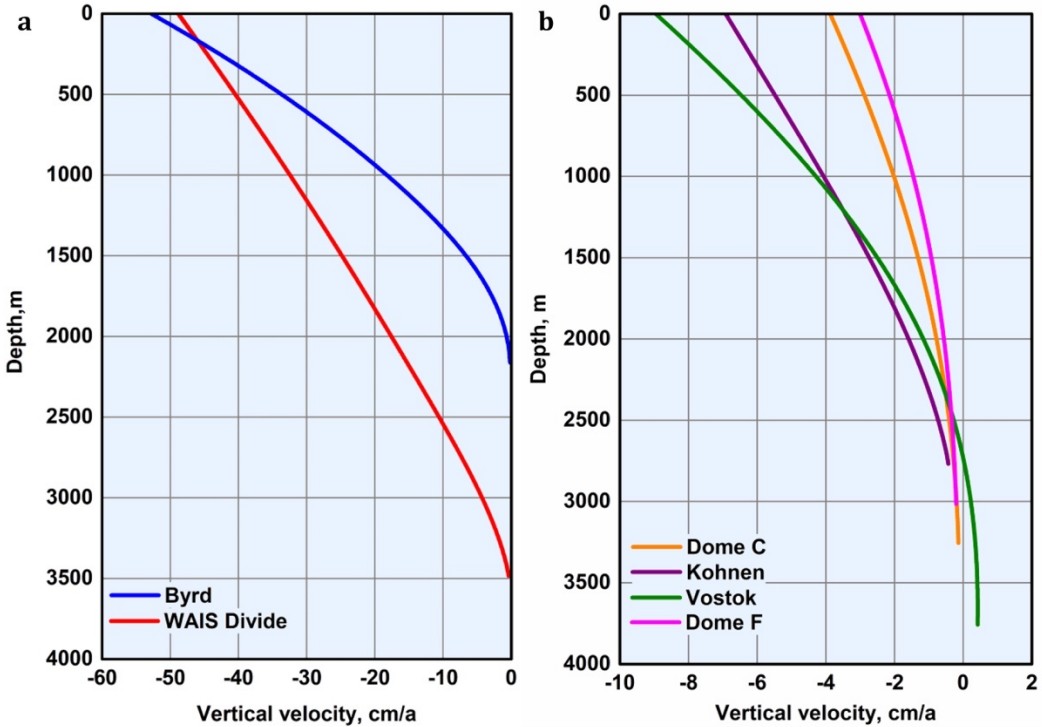

**Figure 5. Estimated vertical velocities at drilling sites in West Antarctica (a) and East Antarctica (b). In East Antarctica snow accumulation and thus vertical ice velocities are far less than in West Antarctica.**

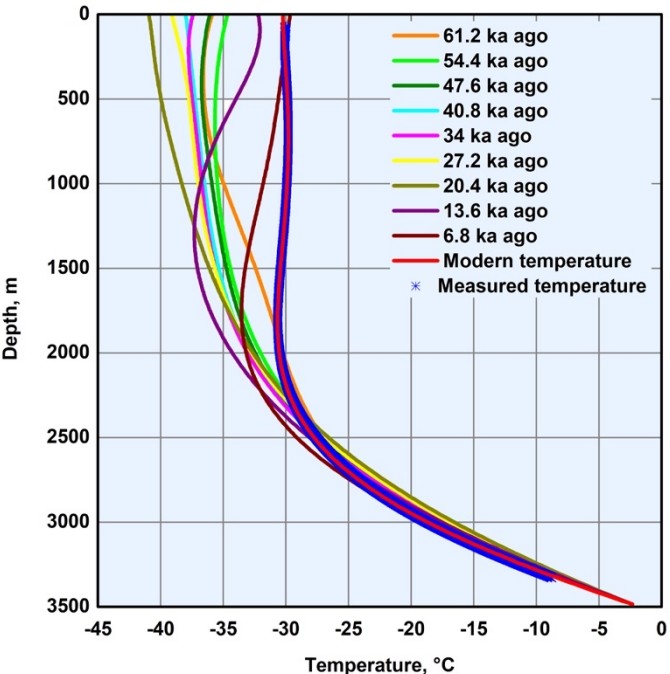


**Figure 6. Paleo temperature profiles at WAIS Divide based on transient model ($m = 0$).**