# Peer review of "Geothermal heat flux from measured temperature profiles in deep ice boreholes in Antarctica"

_The Cryosphere, 2020_

## Referee Comment (RC1) · Anonymous Referee #1 · 14 Apr 2020

The study titled "Geothermal flux beneath the Antarctic Ice Sheet derived from measured temperature profiles in deep boreholes" aims to infer the Geothermal heat flux (GHF) from ice core borehole temperature profiles and heat flow modelling at six ice core sites: Byrd, Dome C, Dome F, Vostok, WAIS Divide, and Kohnen. Talalay et al. apply a thermodynamic model to simulate the temperature of the ice at depth directly against borehole temperature observations to constrain model parameters. A thermodynamic 1D heat flow model was used to infer the GHF at the base of the ice. A least square data-model score of the borehole temperature profiles is performed given four model parameters: surface temperature, surface accumulation rate, basal melt, and basal temperature gradient. A genetic algorithm (GA) is applied to find optimal parameter choices and to infer GHF at the base of each ice core site. The study identified anomalously high GHF values at Kohnen and WAIS Divide relative to the literature.

The study targets pertinent scientific questions with respect to the Antarctic basal environment which are within the scope of TC. However, given issues listed below with the experimental design, the claims of the study are not adequately substantiated and require additional development and experiments.

The title and abstract summarize and reflect the content of the manuscript. The paper follows a logical structure; however, the method section lacks crucial information needed to assess the results of this study. Furthermore, the study does not justify and test their assumptions adequately which directly impact the interpretation of their results against the broader literature (see Main Remarks). Finally, the study does not present information that is required if other researchers were interested in reproducing their work. For these reasons, I suggest that the study is rejected given there are several major revisions required to address the outstanding issues discussed below.

Main Remarks:

1. Heat flow model assumptions

The Antarctic ice sheet has an exceedingly long thermal memory and the slowest response time of the ice sheet is on a timescale exceeding 10 kyr (Ackert, 2003). The ice sheet is continuously in a transient state responding to past changes as well as contemporary forcings. The ice sheet is in disequilibrium, therefore, the assumption that the system is in a thermodynamical steady state must be properly justified and quantified. Otherwise, how are the results of this study meant to be interpreted against the literature (e.g. Martos et al., 2017; Passalacqua et al., 2017).

Over the last several glacial cycles, ice thickness, surface temperatures, and accumulation rates have varied across the ice core sites. Within the scope of a 1D time-dependent heat flow model, these boundary conditions (BCs) directly impact the thermal profile of the ice. Many ice core records offer reconstructions of both temperature and accumulation rates through time. These could directly be applied as BCs into a time-dependent heat flow model rather then constant model parameters.

The structural uncertainty affiliated with the assumption of a steady state heat flow model should be quantified. Time-dependent transient experiments should be conducted with proper time-dependent BCs wherever appropriate to assess the impact of a steady state assumption on the GHF results. Supplemented with a proper uncertainty analysis, this would contextualize the results with the literature.

2. Surface forcing of heat flow model

The heat flow model uses four model parameters: surface temperature, surface accumulation rate, basal melt, and basal temperature gradient. This seems to suggest that the surface temperature and accumulation rate are constant values and not time dependent. What are the resultant optimal GA temperature and accumulation forcings for each ice core site and how do they compare to present day observed values? There is a passing mention of the accumulation rate being time dependent in Section 2.2 to calculate vertical velocities at each ice core site. What is this study using, constant surface accumulation rates (model parameter), time-dependent accumulation rates (vertical velocity inference), or both? How does the accumulation rate used in the vertical velocity calculations compare against the optimal rate inferred from the GA? The study should be consistently using time-dependent surface temperature and accumulation rates. No reference is provided for the accumulation time-series mentioned at line 99, rendering this work not reproducible by other researchers.

3. Understated uncertainties

The GHF results come with uncertainty estimates that only represent one source of uncertainty affiliated with the initial parameter choices going into the GA. This significant underrepresents the overall uncertainties in their GHF estimates, which compromises the interpretation of their results with respect to the literature. The study does not account for structural uncertainties associated with their assumptions (steady state and no horizontal advection). Moreover, it is unclear if the ice thickness in the analysis is kept constant at present day values, this is not explicitly state. It appears the study uses constant ice thickness at each ice core site and does not attempt to estimate GHF uncertainties affiliated with this assumption. The heat flow model does not apply time-dependent surface temperature and accumulation rates, these time-series come with uncertainties which should also be propagated into the uncertainty model of the GHF estimations.

Furthermore, the uncertainty of the power law exponent (form factor) for the vertical velocity profile from Fischer et al. (2013) is not considered. The form factor could be anywhere from m = 0.5 to 1, with the former being favoured by Fischer et al. (2013). The study chooses m=1 without justifying that choice. The analysis should be conducted again using m=0.5 and 0.75 to quantify the impact of the form factor on the GHF estimates. This would propagate parametric uncertainties of the vertical velocity parametrization to the GHF estimates.

The GA manages to identify parameter choices that produce a strong fit to the observed borehole temperatures. However, given the unquantified impact of model assumptions and model weaknesses, it is possible the model is overfitting the data. Therefore, the study would greatly benefit from more robust confidence intervals that incorporate parametric uncertainties and structural errors in the assumptions made in the heat flow model. Upon achieving this, the study would be able to assess the robustness of the anomalous GHF values at Kohnen and WAIS Divide.

Minor comments:

In Figure 1. a GHF comparison is shown at each ice core site. A legend showing which reference is affiliated with which color would clean up the figure and caption. This would remove all the subscript a-e appended onto each GHF bar graph.

References:
Ackert, R. P. (2003) 'An ice sheet remembers', Science (New York, N.Y.), 299(5603), pp. 57–58. doi: 10.1126/science.1079568.

Fischer, H. et al. (2013) 'Where to find 1.5 million yr old ice for the IPICS "Oldest-Ice" ice core', Climate of the Past. doi: 10.5194/cp-9-2489-2013.

Martos, Y. M. et al. (2017) 'Heat Flux Distribution of Antarctica Unveiled', Geophysical Research Letters, 44(22), pp. 11,417-11,426. doi: 10.1002/2017GL075609.

Passalacqua, O. et al. (2017) 'Geothermal flux and basal melt rate in the Dome C region inferred from radar reflectivity and heat modelling', Cryosphere, 11(5), pp. 2231–2246. doi: 10.5194/tc-11-2231-2017.

---

## Referee Comment (RC2) · Anonymous Referee #2 · 18 Jun 2020

In this study the authors provide new estimates of geothermal heat flow (GHF) at six locations in Antarctica: Byrd, WAIS Divide, Dome C, Kohnen, Dome F, and Vostok. The thermal conditions at the base of the ice sheet in the drilling locations as well as the local GHF values are derived through thermodynamic modeling. The GHF is inferred solving the 1D heat-transfer equation in steady state conditions. The results indicate that higher GHF values, with respect to previous studies, are found for two of the locations, Kohnen and WAIS Divide.

The study is within the scope of the Journal and of high interest to the scientific com-

munity. However, there are crucial aspects that are unclear from the text such as, why the results are important, what is the new gained knowledge, how these results compared with other local GHF values obtained through modeling in the same drill sites by other authors? The manuscript lacks of a proper discussion section. In addition, key components of the methods are not adequately described or are missing. In particular, uncertainties are not adequately addressed which makes it difficult to evaluate the results and conclusions of this study.

Below are my comments, suggestions and concerns that I hope will be useful for the authors to improve the manuscript:

- I suggest to change the title as it is not accurately representing the content of the manuscript.

- Regarding the discrepancy between the high values obtained in Kohnen and WAIS Divide in comparison with Antarctic-wide maps:

One thing to consider is that the Antarctic-wide geothermal heat flow maps are representing the heat flow of a region, while a heat flow value derived using borehole measurements is representing a specific local value. Therefore, probably these higher than predicted heat flow values obtained for Kohnen and WAIS Divide are only representing local values, not necessarily hot spots. The higher values could be consequence of, for example, a higher concentration of a particular radiogenic material in that spot, or a consequence of some particularity of the subglacial topography or the parameters and assumptions that are involved in the solutions of the model to obtain the local value. For these reasons, understanding the uncertainty sources and quantifying them is extremely important and it is necessary.

- L69: The manuscript should demonstrate the temperature measurement precision in a robust and scientific way

- L78-80: Where is this shown? Quantify the good agreement. This is important for the

uncertainties of the estimated local geothermal heat flow

- Figure 1: The drill sites as well as other local values are plotted in this figure together with a geological map for the Antarctic continent. However, the geology is not mentioned in the text, there is no discussion about results and the subglacial geology. What is the purpose of the geological map if it is not used in the manuscript? I recommend to either include some discussion about it or select another background data to plot the drill sites and discuss the results in that context.

- Regarding uncertainties I have two main comments/concerns:

1. How uncertainties are calculated is not adequately explained and more information and details are needed to evaluate the GHF estimates.

2. A substantial discussion about which parameters are contributing to the uncertainty is necessary. In addition, there are assumptions made in the thermodynamic model and also parameters that are assumed to be constant. These assumptions also carry uncertainties and they need to be properly quantified and included in the final uncertainty budget. For example, one important aspect to quantify would be the contribution to the uncertainty budget of considering steady-state condition.

- The manuscript should separate results from discussion and conclusions. Additionally, a more detailed discussion is necessary.

---

## Author Comment (AC1) · 29 Jul 2020

**Response to reviewers' comments on the manuscript "Geothermal flux beneath the Antarctic Ice Sheet derived from measured temperature profiles in deep boreholes" submitted to The Cryosphere**

First of all, we would like to thank the Editor Alex Robinson and both anonymous reviewers for fruitful comments and advices. We tried to consider all mentioned issues and, in order to address comments (only critical ones), you will find here our answers point-by-point. The comments are in brown, and our answers are in black. Because our revised manuscript should not be prepared at this stage, we do not present text edits here. After addressing the issues raised, we believe that the manuscript can be accepted by Editor for further processing.

**Anonymous Referee #1**

1. Heat flow model assumptions

The Antarctic ice sheet has an exceedingly long thermal memory and the slowest response time of the ice sheet is on a timescale exceeding 10 kyr (Ackert, 2003). The ice sheet is continuously in a transient state responding to past changes as well as contemporary forcings. The ice sheet is in disequilibrium, therefore, the assumption that the system is in a thermodynamical steady state must be properly justified and quantified. Otherwise, how are the results of this study meant to be interpreted against the literature (e.g. Martos et al., 2017; Passalacqua et al., 2017).

Over the last several glacial cycles, ice thickness, surface temperatures, and accumulation rates have varied across the ice core sites. Within the scope of a 1D time dependent heat flow model, these boundary conditions (BCs) directly impact the thermal profile of the ice. Many ice core records offer reconstructions of both temperature and accumulation rates through time. These could directly be applied as BCs into a time-dependent heat flow model rather then constant model parameters.

The structural uncertainty affiliated with the assumption of a steady state heat flow model should be quantified. Time-dependent transient experiments should be conducted with proper time-dependent BCs wherever appropriate to assess the impact of a steady state assumption on the GHF results. Supplemented with a proper uncertainty analysis, this would contextualize the results with the literature.

We agree with the reviewer that an exact steady state never occurs in reality and transient model likely would give more precise results instead of steady-state model. It is important to recognize that in both cases these will GHF "estimates", not "measurements". The thermal gradient can be affected by processes other than GHF, creating local anomalies that may coincide with the point estimate (e.g., Lake Whillans – Fisher et al., 2015). To use transient model, we need to know the accumulation rate and surface temperature in the past. For some of the discussed drill sites this data is available from ice-core studies, for some sites is not.

To evaluate possibility of using transient model, we did some more calculations for WAIS Divide site. The accumulation rate and surface temperature in the past were taken from the study of Buizert et al., 2015 (Fig. 1). In calculations, the history of ice sheet in WAIS Divide was assumed to be 68000 years long. The governing equation for transient model was solved by Finite Difference Method (FDM). The equation was discretized by both central difference method and upwind difference method and then solved in Matlab. To find best solution, the genetic algorithm (GA) algorithm was still used. The central difference method and upwind difference method demonstrated the same temperature profile. So, here we present the calculation results obtained via upwind difference method.

[Figure]

**Fig. 1.** WAIS Divide ice-core study implications: (a) past temperatures reconstructed from water $\delta$D, calibrated to the borehole temperature profile; (b) past accumulation rates as reconstructed by the firn densification inverse model (red), and from the annual-layer count (black) (Buizert et al., 2015)

Unfortunately, the calculation results with transient model showed the best fit GHF value of $\sim$500 mW m$^{-2}$ when $m = 1$, which seems to be unrealistic. In addition, after running the model, we found that after about 4-8 ka, the influence of initial temperature on temperature profile can be ignored. Later, we assumed the vertical velocity factor $m = 0$ and a GHF value of 235 mW m$^{-2}$ showed good fit with measured temperature (which is close to our estimations of 251.3$\pm$24.1 mW m$^{-2}$ with steady state model and $m = 1$). The temperature distribution in history was modeled 61.2 ka, 54.4ka, … 6.8 ka ago (Fig. 2). As expected, the modelled temperature in the upper part of the ice sheet grossly changes with time but in the lower portion ($\sim$1000 m above ice sheet base) these variations are much smaller. This means the heat disturbance (atmosphere forcing: temperature and precipitation) from the ice sheet surface are gradually decayed with the depth. From all appearances, near-base portion is close to the steady state.

[Figure]

**Fig. 2.** Paleo temperature profiles based on transient model ($m = 0$)

Four drill sites (WAIS Divide, Dome C, Dome F, Vostok) are in close vicinity to ice divides
(Fig. 3a) where horizontal advection and horizontal heat conduction are assumed to be
minimal and the environment approximates a steady state (Cuffey and Paterson, 2010).
In areas with a relatively smooth bed, horizontal conduction is much lower than vertical
conduction (Hindmarsh, 1999, 2018) and horizontal advection and horizontal heat
conduction can also be safely neglected (Van Liefferinge et al., 2018). Therefore, the
interior slow-moving areas of the Antarctic Ice Sheet with smooth bed, including the rest
two sites – Byrd and Kohnen (Fig. 3b), also can be considered at thermal steady state in
their near-base portion.

[Figure]

**Fig. 3.** (a) Antarctic surface ice velocity derived from satellite radar interferometry (Rignot et al., 2011);
(b) Antarctic bed topography (Fretwell et al., 2013) and locations of the deep ice-coring drill sites

Steady state model was intensively used in the past and is still used in the recent GHF
estimates in Antarctica (Martin and Gudmundsson, 2012; Mony et al., 2020; Parrenin et
al., 2017; Price et al., 2002; Zagorodnov et al., 2012 and others). Thus, we are of the
opinion that at first approximation, we can use a steady state model for GHF estimations.
In our method, the temperature in the lower portion of the ice sheets is assumed in steady
state and can be only well fitted by guessing the four key parameters (the surface
temperature, surface accumulation rate, basal melt, and basal temperature gradient).
Utilizing of the GA algorithm helps to find the better value of the four parameters. In GA,
the four parameters are not constrained in the range of the values they appeared in
history because any constraint of the parameters in historical range will show worse
fitting.
It is worth to be mentioned that the value of the parameters for best fitting got by GA is
not the real parameters in ice sheet history. They can be considered as "equivalent"
values for calculating modern temperature profile by eliminating the historical climate
change. Namely, the following steady state equation (Eq. 6 in text) is used to describe only
the temperature profile form instead of calculating the real value of these four
parameters. Consequently, the vertical velocity profile showed in the text is the
"equivalent" vertical velocity instead of real one.

$$T = T_s - \left[\frac{\partial T}{\partial z}\right]_B \int_0^z exp\left(\frac{(w_{melt}-Acc)z^{m+2}}{\alpha_T(m+2)H^{m+1}} - \frac{w_{melt}}{\alpha_T}z\right)dz + \left[\frac{\partial T}{\partial z}\right]_B \int_0^H exp\left(\frac{(w_{melt}-Acc)z^{m+2}}{\alpha_T(m+2)H^{m+1}} - \frac{w_{melt}}{\alpha_T}z\right)dz. \quad (6)$$

The real melt rate is only calculated by the Eq. 7 in text. So, in total, the governing equation
for steady state model was used twice. At the first time, it was integrated to a temperature
distribution form (Eq. 6 in text) to fit lower portion of measured temperature in ice sheets,
and later, the equation was rearranged to calculate real melt rate by derivation of the Eq.
6.

$$w(z) = \left[\alpha_T\left(\frac{\partial^2 T}{\partial z^2}\right) - \alpha_T\left(\frac{1}{k}\frac{\partial k}{\partial T}\right)\left(\frac{\partial T}{\partial z}\right)^2\right] / \frac{\partial T}{\partial z}. \quad (7)$$

So, the only uncertainties in our fitting model are coming from GA algorithm and from
variability of the form factor $m$. For each deep borehole, the fitting experiments were
trialed five times to avoid random error of GA. Then, the average value in the five fitting
experiments was used as the GHF from bedrock into ice at selected site and the
uncertainty ranges came from the difference between the maximum/minimum and the
average GHF values.

Choosing of the appropriate form factor $m$ is really challenging task. In our manuscript
we stated (Lines 101-105):

"Classically, vertical velocity linearly depends on $z/H$ (Cuffey and Paterson, 2010) and $m$
= 0. However, at an ice divide, the downward flow of ice is slower, for the same depth,
than at locations away from the divide (Raymond, 1983). This reduces the cooling
influence of vertical advection and increases the basal temperature. Within this near-
divide zone, the form factor could be from 0.5 (Fischer et al., 2013) to 1.0 (Raymond,
1983). All discussed sites are located at, or near, ice divide, thus, we assume $m$ = 1."

To set up the vertical velocity profile at Dome C, Fischer et al. (2013) performed three
runs with $m$ = 0.3, $m$ = 0.5 and $m$ = 0.7 and found that the temperature profile is only
slightly affected by this choice. However, the form factor $m$ had a strong influence on the
age profile of the ice. That was the reason why the authors used $m$ = 0.5, which is in good
agreement with the EDC3 age scale.

To evaluate influence of $m$ in the steady state model, we did some more calculations again
for WAIS Divide site. The results are summarized in the table below. The results from the
previous estimations that were trialed five times with $m$ = 1 are highlighted by grey.

| $m$ | 1 | 1 | 1 | 1 | 1 | Average value | Error | 0,75 | 0,5 | 0,25 | 0 |
|---|---|---|---|---|---|---|---|---|---|---|---|
| Interval, m | 2000-bottom | 2000-bottom | 2000-bottom | 2000-bottom | 2000-bottom | | | 2000-bottom | 2000-bottom | 2000-bottom | 2000-bottom |
| Temperature gradient (℃/100m) | 3,98 | 3,75 | 3,9 | 3,92 | 3,87 | 3,88 | | 3,93 | 3,71 | 3,59 | 3,69 |
| Melt rate (cm/a) | 1,97 | 1,52 | 1,85 | 1,82 | 1,78 | 1,75 | 0,23 | 1,62 | 1,30 | 0,24 | -0,05 |
| Conductive flux (mW/m^2) | 83,7 | 79,1 | 82,1 | 82,6 | 81,5 | 81,4 | 2,3 | 82,78 | 81,55 | 75,86 | 77,75 |
| Melt flux (mW/m^2) | 191,7 | 148 | 179,5 | 176,3 | 172,9 | 169,9 | 21,9 | 157,56 | 125,87 | 24,22 | -4,88 |
| GHF (mW/m^2) | 275,4 | 227,2 | 261,7 | 259 | 254,5 | 251,3 | 24,1 | 240,3 | 207,4 | 100 | 72,8 |

Decreasing of $m$ from 0.75 to 0 leads to reduction of GHF from 240.3 to 72.8 mW m$^{-2}$. In
the latter case, the melt flux is negative meaning that the ice sheet base is frozen to the
bed (it is extremely unlikely). Because there are no clear considerations or prerequisites
for choosing of the form factor $m$, we still are on the opinion that at the first approximation we can follow Raymond's (1983) arguments for deformation in the
vicinity of ice divides and run with $m = 1$.

From the other hand, GHFs for WAIS Divide and Kohnen look really overestimated. The
current WAIS team modeling shows that GHF at WAIS Divide is on the order of 105 mW
$m^{-2}$ (unpublished as of yet) that corresponds to our GA result with steady state model and
form factor $m = 0.25$ (~100 mW $m^{-2}$). Apparently, there are some other "physical"
uncertainties that we did not know and did not account for these sites at the moment.
Most likely that horizontal conduction in the bottom of the ice sheet at these sites is quite
high and cannot be ignored. Thus, one of co-authors suggested to remove results of our
GHF estimates for WAIS Divide and Kohnen from the revised version of the manuscript
as some unsolved problems of the applied model and GA are still exist. However, we can
try to go further with GHF estimates with GA for other four sites. Our further actions will
depend on feedback from the editor, reviewers and final discussion within co-authors.

2. Surface forcing of heat flow model

The heat flow model uses four model parameters: surface temperature, surface
accumulation rate, basal melt, and basal temperature gradient. This seems to suggest that
the surface temperature and accumulation rate are constant values and not time
dependent. What are the resultant optimal GA temperature and accumulation forcings
for each ice core site and how do they compare to present day observed values? There is
a passing mention of the accumulation rate being time dependent in Section 2.2 to
calculate vertical velocities at each ice core site. What is this study using, constant surface
accumulation rates (model parameter), time-dependent accumulation rates (vertical
velocity inference), or both? How does the accumulation rate used in the vertical velocity
calculations compare against the optimal rate inferred from the GA? The study should be
consistently using time-dependent surface temperature and accumulation rates. No
reference is provided for the accumulation time-series mentioned at line 99, rendering
this work not reproducible by other researchers.

Section 2.2 of submitted manuscript contains general theoretical computation of
temperature distribution in ice sheet at steady state. In our calculations, we do not use
time-dependent values of the surface temperature and accumulation rate. These
parameters are changed within GA in a wide range to fit the measured temperature. Then
the "equivalent" accumulation rates and temperature can be calculated from GA results.
The following table shows values of "equivalent" accumulation rates and temperature at
ice sheet surface which were derived from our calculations. In all cases, "equivalent"
accumulation rates are higher than modern ones while the "equivalent" surface
temperature are close to modern ones. "Equivalent" vertical velocities at drilling sites
derived from GA results are shown on Fig. 3 of submitted manuscript.

| Parameters | Byrd | WAIS Divide | Vostok | Dome C | Kohnen | Dome F |
|---|---|---|---|---|---|---|
| "Equivalent" Acc (cm $a^{-1}$) | 64.7 | 97.4 | 8.95 | 3.70 | 7.22 | 3.00 |
| Modern Acc (cm $a^{-1}$) | 16.9 | 22.0 | 2.48 | 2.84 | 7.00 | 2.99 |
| "Equivalent" surface temperature (°C) | -29 | -30.8 | -56.5 | -54.6 | -45 | -56.5 |
| Modern surface temperature (°C) | -28 | -30 | -57 | -54.6 | -44 | -57.3 |

In general, the computational details that need to be captured and shared for
reproducible research include: (1) the data that were used in the analysis; (2) written statements in a programming language (i.e., the source code of the software used in the
analysis or to generate data products); (3) numeric values of all configurable settings for
software; (4) detailed specification of computational environment including system
software and hardware requirements, including the version number of each software
used; and (5) computational workflow (National Academies of Sciences, Engineering, and
Medicine, 2019). All these are extremely extensive. We tried to provide baseline that can
guarantee reproducibility of our scientific findings and will be happy to provide other
data (if considered necessary).

3. Understated uncertainties
The GHF results come with uncertainty estimates that only represent one source of
uncertainty affiliated with the initial parameter choices going into the GA. This significant
underrepresents the overall uncertainties in their GHF estimates, which compromises
the interpretation of their results with respect to the literature. The study does not
account for structural uncertainties associated with their assumptions (steady state and
no horizontal advection). Moreover, it is unclear if the ice thickness in the analysis is kept
constant at present day values, this is not explicitly state. It appears the study uses
constant ice thickness at each ice core site and does not attempt to estimate GHF
uncertainties affiliated with this assumption. The heat flow model does not apply time-
dependent surface temperature and accumulation rates, these time-series come with
uncertainties which should also be propagated into the uncertainty model of the GHF
estimations.
Furthermore, the uncertainty of the power law exponent (form factor) for the vertical
velocity profile from Fischer et al. (2013) is not considered. The form factor could be
anywhere from m = 0.5 to 1, with the former being favoured by Fischer et al. (2013). The
study chooses m=1 without justifying that choice. The analysis should be conducted again
using m=0.5 and 0.75 to quantify the impact of the form factor on the GHF estimates. This
would propagate parametric uncertainties of the vertical velocity parametrization to the
GHF estimates.
The GA manages to identify parameter choices that produce a strong fit to the observed
borehole temperatures. However, given the unquantified impact of model assumptions
and model weaknesses, it is possible the model is overfitting the data. Therefore, the
study would greatly benefit from more robust confidence intervals that incorporate
parametric uncertainties and structural errors in the assumptions made in the heat flow
model. Upon achieving this, the study would be able to assess the robustness of the
anomalous GHF values at Kohnen and WAIS Divide.
In our calculations, the ice sheet thickness was kept at present day height. We agree with
the reviewer that the ice sheet thickness at the studied sites varied in history. 3D thermo-
mechanical model and the simple 1D model showed that the maximum variation of ice
sheet thickness at Dome C and Dome F is less than 250 m in the history (Parrenin et al.,
2007). Generally, the typical difference in the ice thickness in the glacial and interglacial
periods at Dome C was 150 m (Passalacqua et al., 2017). At the Kohnen site, the local
elevation variation is in the order of 100 m (Huybrechts et al., 2007). Located in east
Antarctica plateau, the ice thickness variation at Vostok has the similar range as at Dome
F and Dome C (Ritz et al., 2001).
The best evidences for ice-sheet elevation change in the interior of the West Antarctic ice
sheet come from the Ohio Range, to the south of the WAIS Divide site at a height of 1600
m a.s.l., and from Mt. Waesche to the north of the WAIS Divide site at a height of 2000 m a.s.l. (Ackert et al., 1999, 2007). Moraines at Mt. Waesche were ~50 m higher and
trimlines in the Ohio Range were ~125 m higher, between 12 and 10 ka. The ~100 m of
thinning throughout the Holocene occurred as the grounding line retreated by hundreds
of km and the accumulation rates were relatively stable (Anderson et al., 2002; Conway
et al., 1999). The model of Cuffey et al. (2016) suggested that a more likely scenario of
200 m thickening at WAIS Divide when accumulation rises after the last glacial maximum,
followed by 300 m of thinning to the mid-Holocene. The elevation change is comparable
to the amount of elevation change inferred for interior East Antarctic sites.

Comparing with the modern ice thickness value, the variation of ice thickness is small and
its influence on ice temperature distribution can be neglected, especially on lower
portion of the ice borehole. For example, assuming a 150 m thickness increase from the
LGM to 15 ka changes the reconstructed LGM temperature by less than 0.2 °C compared
to a constant thickness in WAIS ice core (Buizert et al., 2015).  This is the reason why
constant ice thickness is also used by other researchers for GHF estimations (Dahl-Jensen
et al., 2003; Engelhardt, 2004; Mony et al., 2020).

To our knowledge, we used GA for the first time to find optimal solution of temperature
fitting in ice sheets. GA can solve optimization problems by limiting unknown parameters
changing in a predetermined range with any types of constraints, including integer
constraints. This is very helpful in our study. In general, GA generates high-quality
solutions for optimization problems and search problems.

Answers on the other comments were given hereinbefore.

Minor comments:

In Figure 1. a GHF comparison is shown at each ice core site. A legend showing which
reference is affiliated with which color would clean up the figure and caption. This would
remove all the subscript a-e appended onto each GHF bar graph.

We will be happy to add corrected figure in the revised version of the manuscript (see
Fig. 4 in the current response).

**Anonymous Referee #2**

… there are crucial aspects that are unclear from the text such as, why the results are
important, what is the new gained knowledge, how these results compared with other
local GHF values obtained through modeling in the same drill sites by other authors?

The Antarctic GHF is an important boundary condition for ice sheet behavior and
associated sea level change (Golledge et al., 2015) since it keeps basal ice relatively warm,
and thus less viscous than colder ice above, and helps supply meltwater at the ice sheet
base. Typical questions are: What are the basal ice temperature and mechanical
properties? How does GHF control basal melt and internal deformation of the ice sheet?
How old is ice at different locations? These questions can be answered only by applying
reliable GHF measurements or estimates. However, GHF remains poorly constrained,
with few borehole-derived estimates, and there are large discrepancies in currently
available glaciological and geophysical estimates (Burton-Johnson et al., 2020).

We estimated GHF at six sites – Byrd, WAIS Divide, Dome C, Kohnen, Dome F, and Vostok,
which have succeeded in reaching to, or nearly to, the bed in inland locations in Antarctica.
Our GHF estimates allow to validate continental and local models and reveal (if so) local
geothermal anomalies.

Obtained GHF values are compared with five modellings (Shapiro and Ritzwoller, 2004; Fox Maule et al., 2005 Van Liefferinge and Pattyn, 2013; An et al., 2015; Martos et al., 2017) using bar graphs on the Fig. 1 (or Fig. 4 in the current response). Further comparison with this data and data from other references for specific sites was given in the "Results and discussion" section.

The manuscript lacks of a proper discussion section. The manuscript should separate results from discussion and conclusions. Additionally, a more detailed discussion is necessary.

In case if the manuscript will be accepted by Editor for further processing, we separate "Results and discussion" section into two sections "Results" and "Discussion". We plan to add into "Discussion" the following details: (1) transient model vs. steady state model; (2) uncertainties; (3) comparison studies; (4) implications. In addition, "Conclusions" section will be presented.

In addition, key components of the methods are not adequately described or are missing. In particular, uncertainties are not adequately addressed which makes it difficult to evaluate the results and conclusions of this study.

We applied GA algorithm to fit measured temperature in deep ice-core drilling boreholes by variation guessing of the four key parameters influenced on temperature distribution: the surface temperature, surface accumulation rate, basal melt, and basal temperature gradient. All these parameters are suggested by algorithm in order to get the best-fitting curve. The only uncertainties in our fitting model are coming from GA algorithm itself and from variability of the form factor $m$. Detailed answers for applied method and uncertainties are given to Anonymous Referee #1 hereinbefore.

Below are my comments, suggestions and concerns that I hope will be useful for the authors to improve the manuscript:
- I suggest to change the title as it is not accurately representing the content of the manuscript.

The title can be changed to: "Geothermal heat flux from temperature profiles in deep ice-core drilling boreholes in Antarctica".

- Regarding the discrepancy between the high values obtained in Kohnen and WAIS Divide in comparison with Antarctic-wide maps:
One thing to consider is that the Antarctic-wide geothermal heat flow maps are representing the heat flow of a region, while a heat flow value derived using borehole measurements is representing a specific local value. Therefore, probably these higher than predicted heat flow values obtained for Kohnen and WAIS Divide are only representing local values, not necessarily hot spots. The higher values could be consequence of, for example, a higher concentration of a particular radiogenic material in that spot, or a consequence of some particularity of the subglacial topography or the parameters and assumptions that are involved in the solutions of the model to obtain the local value. For these reasons, understanding the uncertainty sources and quantifying them is extremely important and it is necessary.

We agree with the reviewer that there are a large number of possible reasons for elevated GHF at Kohnen and WAIS Divide (variability of crustal thickness, hydrothermal circulation, high concentration of radiogenic materials in the bedrock, etc.). However, at this stage we are not able to give lucid explanations of this phenomenon. Uncertainty sources and quantifying were discussed above.

- L69: The manuscript should demonstrate the temperature measurement precision in a robust and scientific way

We will add evaluation of temperature measurement precision in the revised version of the paper. Uncertainties of temperature measurements in mechanically drilled boreholes filled with drilling fluids were presented in details by one of co-authors in USGS report (Clow, 2008).

- L78-80: Where is this shown? Quantify the good agreement. This is important for the uncertainties of the estimated local geothermal heat flow

This is shown in Table 1 (the line "Ice thickness according with radar/seismic survey (m)") and Table 2 (the line "Ice thickness according with depth of pressure melting point (m)").

- Figure 1: The drill sites as well as other local values are plotted in this figure together with a geological map for the Antarctic continent. However, the geology is not mentioned in the text, there is no discussion about results and the subglacial geology. What is the purpose of the geological map if it is not used in the manuscript? I recommend to either include some discussion about it or select another background data to plot the drill sites and discuss the results in that context.

At the first stage of the paper writing, we planned to connect revealed GHF values with Antarctic subglacial geology but then, because of the insufficiency of data, we dropped this idea. We agree with reviewer that it would be more rational to select another background, for example, with location of the Antarctic ice divides (Fig. 4).

- Regarding uncertainties I have two main comments/concerns:

1. How uncertainties are calculated is not adequately explained and more information and details are needed to evaluate the GHF estimates.

2. A substantial discussion about which parameters are contributing to the uncertainty is necessary. In addition, there are assumptions made in the thermodynamic model and also parameters that are assumed to be constant. These assumptions also carry uncertainties and they need to be properly quantified and included in the final uncertainty budget. For example, one important aspect to quantify would be the contribution to the uncertainty budget of considering steady-state condition.

We will be happy to add uncertainty considerations into revised version of the paper.

**References**

Ackert RP, David JB, Harold WB, Parker EC. Mark DK, James LF, Eric J (1999). Measurements of past ice sheet elevations in interior West Antarctica. Science 286(5438), 276-280.

Ackert RP, Mukhopadhyay S, Parizek BR, and Borns HW (2007). Ice elevation near the West Antarctic Ice Sheet divide during the Last Glaciation, Geophys. Res. Lett., 34 (21), L21506.Anderson JB, Shipp SS, Lowe AL, Wellner JS, Mosola AB (2002). The Antarctic Ice

Sheet during the Last Glacial Maximum and its subsequent retreat history: a review. Quaternary Science Reviews, 21, 49-70.

[Figure]

**Fig. 4.** GHF derived in the present study (P.S.) from basal temperature gradients in deep ice boreholes (green bars) compared with modelling. Location of the Antarctic ice divides is shown according with Rignot et al., 2011.

An M, Wiens DA, Zhao Y, Feng M, Nyblade A, Kanao M, Li Y, Maggi A, Lévêque J-J (2015) Temperature, lithosphere-asthenosphere boundary, and heat flux beneath the Antarctic Plate inferred from seismic velocities. Journal of Geophysical Research: Solid Earth, 120, 8720–8742.

Buizert C, Cuffey KM, Severinghaus JP, Baggenstos D, Fudge TJ, Steig EJ, Markle BR, Winstrup M, Rhodes RH, Brook EJ, Sowers TA, Clow GD, Cheng H, Edwards RL, Sigl M, McConnell JR, Taylor KC (2015). The WAIS Divide deep ice core WD2014 chronology – Part 1: Methane synchronization (68–31 ka BP) and the gas age–ice age difference Clim. Past, 11, 153–173.

Burton-Johnson A, Dziadek R, Martin, C. (2020). Geothermal heat flow in Antarctica: current and future directions, The Cryosphere Discuss., in review.

Clow GD (2008) USGS Polar temperature logging system, description and measurement uncertainties: U.S. Geological Survey Techniques and Methods 2–E3.

Cuffey KM, Paterson WSB (2010). The physics of glaciers, 4th edn. Butterworth-Heinemann, Oxford.

Conway H, Hall BL, Denton GH, Gades AM, Waddington ED (1999). Past and future grounding-line retreat of the West Antarctic Ice Sheet. Science, 286, 280-283.

Cuffey KM, Clow GD, Steig EJ, Buizert C, Fudge TJ, Koutnik M, Waddington ED, Alley RB, Severinghaus JP (2016). Deglacial temperature history of West Antarctica. PNAS, 113 (50), 14249–14254.

Dahl-Jensen D, Gundestrup N, Gogineni SP, Miller H (2003). Basal melt at NorthGRIP modeled from borehole, ice-core and radio-echo sounder observations. Ann. Glaciol., 37, 217-212.

Engelhardt H (2004) Ice temperature and high geothermal flux at Siple Dome, West Antarctica, from borehole measurements. J Glaciol., 50(169), 251-256.

Fischer H, Severinghaus J, Brook E, Wolff E, Albert M, Alemany O, Arthern R, Bentley C, Blankenship D, Chappellaz J, Creyts T, Dahl-Jensen D, Dinn M, Frezzotti M, Fujita S, Gallee H, Hindmarsh R, Hudspeth D, Jugie G, Kawamura K, Lipenkov V, Miller H, Mulvaney R, Parrenin F, Pattyn F, Ritz C, Schwander J, Steinhage D, van Ommen T, Wilhelms F (2013). Where to find 1.5 million yr old ice for the IPICS "Oldest-Ice" ice core. Clim. Past, 9, 2489–2505.

Fox Maule CF, Purucker ME, Olsen N, Mosegaard K. (2005) Heat flux anomalies in Antarctica revealed by satellite magnetic data. Science, 309, 464–467.

Fretwell P and 59 others (2013). Bedmap 2: improved ice bed, surface and thickness datasets for Antarctica. The Cryosphere, 7, 375–393.

Golledge NR, Kowalewski DE, Naish TR, Levy RH, Fogwill CJ, Gasson EGW (2015). The multi-millennial Antarctic commitment to future sea-level rise. Nature, 526(7573), 421–425.

Hindmarsh RCA (1999). On the numerical computation of temperature in an ice sheet. J. Glaciol., 45, 568–574.

Hindmarsh R (2018). Ice-sheet and glacier modelling, in: Past Glacial Environments, Elsevier, 605–661.

Huybrechts P, Rybak O, Pattyn F, Ruth U, Steinhage D (2007). Ice thinning, upstream advection, and non-climatic biases for the upper 89% of the EDML ice core from a nested model of the Antarctic ice sheet. Climate of the Past, 3 (4), 577 - 589.

Martin C, Gudmundsson GH (2012). Effects of nonlinear rheology, temperature and anisotropy on the relationship between age and depth at ice divides. The Cryosphere, 6, 1221–1229.

Martos YM, Catalán M, Jordan TA, Golynsky A, Golynsky D, Eagles G, Vaughan DG (2017) Heat flux distribution of Antarctica unveiled. Geophys Res Lett., 44, 11417–11426.

Mony L, Roberts JL, Halpin JA (2020). Inferring geothermal heat flux from an ice-borehole temperature profile at Law Dome, East Antarctica. J. Glaciol. 1–11.

National Academies of Sciences, Engineering, and Medicine (2019). Reproducibility and replicability in science. Washington, DC: The National Academies Press.

Parrenin F, Dreyfus G, Durand G, Fujita S, Gagliardini O, Gillet F, Jouze J, Kawamura K, Lhomme N, Masson-Delmotte V, Ritz C, Schwander J, Shoji H, Uemura R, Watanabe O, Yoshida N (2007). 1-D-ice flow modelling at EPICA Dome C and Dome Fuji, East Antarctica. Climate of the Past, 3 (2), 243 – 259.Parrenin F, Cavitte MGP, Blankenship DD, Chappellaz J, Fischer H, Gagliardini O, Masson-Delmotte V, Passalacqua O, Ritz C, Roberts J, Siegert MJ, Young DA (2017). Is there 1.5-million-year-old ice near Dome C, Antarctica? The Cryosphere, 11, 2427–2437.

Passalacqua O, Ritz C, Parrenin F, Urbini S, Frezzott M (2017). Geothermal flux and basal melt rate in the Dome C region inferred from radar reflectivity and heat modelling. Cryosphere, 11(5), 2231–2246. doi: 10.5194/tc-11-2231-2017.

Price PB, Nagornov OV, Bay R, Chirkin D, He Y, Miocinovic P, Richards A, Woschnagg K, Koci B, Zagorodnov V (2002). Temperature profile for glacial ice at the South Pole: Implications for life in a nearby subglacial lake. PNAS 99(12), 7844–7847.

Raymond CF (1983) Deformation in the vicinity of ice divides. J. Glaciol., 29, 357–373.

Rignot E, Mouginot J, Scheuchl B (2011). Ice flow of the Antarctic Ice Sheet. Science 333, 1427-1430.

Ritz C, Rommelaere V, Dumas C (2001). Modeling the evolution of Antarctic ice sheet over the last
420,000 years: Implications for altitude changes in the Vostok region. Journal of
Geophysical Research: Atmospheres. 106 (D23), 31943 – 31964.

Shapiro NM, Ritzwoller MH (2004) Inferring surface heat flux distributions guided by a global
seismic model: particular application to Antarctica. Earth Planet. Sci. Lett., 223, 213-224.

Van Liefferinge B, Pattyn F (2013) Using ice-flow models to evaluate potential sites of million
year-old ice in Antarctica. Clim. Past, 9, 2335–2345.

Van Liefferinge B, Pattyn F, Cavitte MGP, Karlsson NB, Young DA, Sutter J, Eisen O (2018)
Promising Oldest Ice sites in East Antarctica based on thermodynamical modelling. The
Cryosphere, 12, 2773–2787.

Zagorodnov V, Nagornov O, Scambos TA, Muto A, Mosley-Thompson E, Pettit EC, S. Tyuflin S
(2012). Borehole temperatures reveal details of 20th century warming at Bruce Plateau,
Antarctic Peninsula. The Cryosphere, 6, 675–686.

On behalf of co-authors:

Pavel Talalay

Yazhou Li

---

## Author Response (AR1)

**Response to reviewers' comments on the manuscript "Geothermal flux beneath the Antarctic Ice Sheet derived from measured temperature profiles in deep boreholes" submitted to The Cryosphere**

5 First of all, we would like to thank the Editor Alex Robinson and both anonymous reviewers for fruitful comments and advices. We tried to consider all mentioned issues and, in order to address comments (only critical ones), you will find here our answers point-by-point. The comments are in **brown**, and our answers are in black. The list of all relevant changes made in the manuscript and the marked-up manuscript version are placed at the end of the response letter.

10

**Anonymous Referee #1**

**1. Heat flow model assumptions**

The Antarctic ice sheet has an exceedingly long thermal memory and the slowest response time of the ice sheet is on a timescale exceeding 10 kyr (Ackert, 2003). The ice sheet is continuously in a

15 transient state responding to past changes as well as contemporary forcings. The ice sheet is in disequilibrium, therefore, the assumption that the system is in a thermodynamical steady state must be properly justified and quantified. Otherwise, how are the results of this study meant to be interpreted against the literature (e.g. Martos et al., 2017; Passalacqua et al., 2017).

Over the last several glacial cycles, ice thickness, surface temperatures, and accumulation rates

- 20 have varied across the ice core sites. Within the scope of a 1D time dependent heat flow model, these boundary conditions (BCs) directly impact the thermal profile of the ice. Many ice core records offer reconstructions of both temperature and accumulation rates through time. These could directly be applied as BCs into a time-dependent heat flow model rather then constant model parameters.
- 25 The structural uncertainty affiliated with the assumption of a steady state heat flow model should be quantified. Time-dependent transient experiments should be conducted with proper timedependent BCs wherever appropriate to assess the impact of a steady state assumption on the GHF results. Supplemented with a proper uncertainty analysis, this would contextualize the results with the literature.
- 30 Three drill sites (Dome C, Dome F, Vostok) are in close vicinity to ice divides where horizontal advection and horizontal heat conduction are assumed to be minimal and the environment approximates a steady state (Cuffey and Paterson, 2010). To first order, we also assume WAIS Divide is in a steady state. Byrd and Kohnen are in the interior slow-moving areas of the Antarctic Ice Sheet with a relatively smooth bed, where horizontal conduction is much lower than vertical
- 35 conduction (Hindmarsh, 1999, 2018), as well as horizontal advection and horizontal heat conduction can be neglected (Robin, 1955; Van Liefferinge et al., 2018). Thus, we assume that the temperature measured at the six drill sites is at thermal steady state in the near base portion.

We modified the design method and chose the form factor for concrete site according with the best fitting between modeled and measured depth-age scales. The best value for the form factor was

40 selected on the basis of the nonlinear correlation analysis. We think that the corrected design method gave more exact results. In addition, we added into manuscript section 2.3 "Uncertainties" and section 4.1 "Transient model vs. steady-state model".

**2. Surface forcing of heat flow model**

- 45 The heat flow model uses four model parameters: surface temperature, surface accumulation rate, basal melt, and basal temperature gradient. This seems to suggest that the surface temperature and accumulation rate are constant values and not time dependent. What are the resultant optimal GA temperature and accumulation forcings for each ice core site and how do they compare to present day observed values? There is a passing mention of the accumulation rate being time dependent in
- 50 Section 2.2 to calculate vertical velocities at each ice core site. What is this study using, constant surface accumulation rates (model parameter), time-dependent accumulation rates (vertical velocity inference), or both? How does the accumulation rate used in the vertical velocity calculations compare against the optimal rate inferred from the GA? The study should be consistently using time-dependent surface temperature and accumulation rates. No reference is
- 55 provided for the accumulation time-series mentioned at line 99, rendering this work not reproducible by other researchers.

In our calculations, we do not use time-dependent values of the surface temperature and accumulation rate. These parameters are changed within GA in a wide range to fit the measured temperature. Then "equivalent" vertical velocity, modern accumulation rate and temperature can be calculated from the GA results (please, see section 3.3 "Indirect results").

- 60 be calculated from the GA results (please, see section 3.3 "Indirect results"). In general, the computational details that need to be captured and shared for reproducible research include: (1) the data that were used in the analysis; (2) written statements in a programming language (i.e., the source code of the software used in the analysis or to generate data products); (3) numeric values of all configurable settings for software; (4) detailed specification of computational
- 65 environment including system software and hardware requirements, including the version number of each software used; and (5) computational workflow1. All these are extremely extensive. We tried to provide baseline that can guarantee reproducibility of our scientific findings and will be happy to provide other data (if considered necessary).
- 70 3. Understated uncertainties

The GHF results come with uncertainty estimates that only represent one source of uncertainty affiliated with the initial parameter choices going into the GA. This significant underrepresents the overall uncertainties in their GHF estimates, which compromises the interpretation of their results with respect to the literature. The study does not account for structural uncertainties associated

75 with their assumptions (steady state and no horizontal advection). Moreover, it is unclear if the ice thickness in the analysis is kept constant at present day values, this is not explicitly state. It appears

<sup>1 National Academies of Sciences, Engineering, and Medicine (2019). Reproducibility and replicability in science. Washington, DC: The National Academies Press.

the study uses constant ice thickness at each ice core site and does not attempt to estimate GHF uncertainties affiliated with this assumption. The heat flow model does not apply time-dependent surface temperature and accumulation rates, these time-series come with uncertainties which should also be propagated into the uncertainty model of the GHF estimations.

Furthermore, the uncertainty of the power law exponent (form factor) for the vertical velocity profile from Fischer et al. (2013) is not considered. The form factor could be anywhere from m = 0.5 to 1, with the former being favoured by Fischer et al. (2013). The study chooses m=1 without justifying that choice. The analysis should be conducted again using m=0.5 and 0.75 to quantify the

85 impact of the form factor on the GHF estimates. This would propagate parametric uncertainties of the vertical velocity parametrization to the GHF estimates.

The GA manages to identify parameter choices that produce a strong fit to the observed borehole temperatures. However, given the unquantified impact of model assumptions and model weaknesses, it is possible the model is overfitting the data. Therefore, the study would greatly

90 benefit from more robust confidence intervals that incorporate parametric uncertainties and structural errors in the assumptions made in the heat flow model. Upon achieving this, the study would be able to assess the robustness of the anomalous GHF values at Kohnen and WAIS Divide.

For uncertainty analysis, please, see section 2.3 in the revised manuscript below.

95 Minor comments:

In Figure 1. a GHF comparison is shown at each ice core site. A legend showing which reference is affiliated with which color would clean up the figure and caption. This would remove all the subscript a-e appended onto each GHF bar graph.

Figure 1 is corrected.

100

80

**Anonymous Referee #2**

... there are crucial aspects that are unclear from the text such as, why the results are important, what is the new gained knowledge, how these results compared with other local GHF values obtained through modeling in the same drill sites by other authors?

- 105 The importance of these studies is emphasized at the beginning of "Introduction". Obtained GHF values are compared with five modellings (Shapiro and Ritzwoller, 2004; Fox Maule et al., 2005 Van Liefferinge and Pattyn, 2013; An et al., 2015; Martos et al., 2017) using bar graphs on the Fig. 1. Further comparison with this data and data from other references for specific sites is given in the section 3.2 "Data comparison and divergences".
- 110

The manuscript lacks of a proper discussion section. The manuscript should separate results from discussion and conclusions. Additionally, a more detailed discussion is necessary.

"Discussion" and "Conclusions" sections are added into revised version of the manuscript.

115 In addition, key components of the methods are not adequately described or are missing. In particular, uncertainties are not adequately addressed which makes it difficult to evaluate the results and conclusions of this study.

We added section 2.3 "Uncertainties".

120 Below are my comments, suggestions and concerns that I hope will be useful for the authors to improve the manuscript:

- I suggest to change the title as it is not accurately representing the content of the manuscript.

The title of the manuscript is changed to: "Geothermal heat flux from measured temperature profiles in deep ice boreholes in Antarctica".

125

- Regarding the discrepancy between the high values obtained in Kohnen and WAIS Divide in comparison with Antarctic-wide maps:

One thing to consider is that the Antarctic-wide geothermal heat flow maps are representing the heat flow of a region, while a heat flow value derived using borehole measurements is representing

- 130 a specific local value. Therefore, probably these higher than predicted heat flow values obtained for Kohnen and WAIS Divide are only representing local values, not necessarily hot spots. The higher values could be consequence of, for example, a higher concentration of a particular radiogenic material in that spot, or a consequence of some particularity of the subglacial topography or the parameters and assumptions that are involved in the solutions of the model to obtain the local value.
- 135 For these reasons, understanding the uncertainty sources and quantifying them is extremely important and it is necessary.

We recalculated GHF in Kohnen and WAIS Divide according to the best value for the form factor that was selected on the basis of the nonlinear correlation analysis between modeled and measured age scales.

140

- L69: The manuscript should demonstrate the temperature measurement precision in a robust and scientific way

We added section 2.3.1 "Temperature measurements" with explanations of temperature measurement precision.

145

- L78-80: Where is this shown? Quantify the good agreement. This is important for the uncertainties of the estimated local geothermal heat flow

This is shown in Table 1 (the line "Ice thickness according with radar/seismic survey (m)") and Table 2 (the line "Ice thickness according with depth of pressure melting point (m)").

- Figure 1: The drill sites as well as other local values are plotted in this figure together with a geological map for the Antarctic continent. However, the geology is not mentioned in the text, there is no discussion about results and the subglacial geology. What is the purpose of the geological map if it is not used in the manuscript? I recommend to either include some discussion about it or select another background data to plot the drill sites and discuss the results in that context.

- At the first stage of the paper writing, we planned to connect revealed GHF values with Antarctic subglacial geology but then, because of the insufficiency of data, we dropped this idea. We redrew this figure and added location of the Antarctic ice divides.
- 160 Regarding uncertainties I have two main comments/concerns:

1. How uncertainties are calculated is not adequately explained and more information and details are needed to evaluate the GHF estimates.

2. A substantial discussion about which parameters are contributing to the uncertainty is necessary. In addition, there are assumptions made in the thermodynamic model and also parameters that are

165 assumed to be constant. These assumptions also carry uncertainties and they need to be properly quantified and included in the final uncertainty budget. For example, one important aspect to quantify would be the contribution to the uncertainty budget of considering steady-state condition. We added uncertainty considerations into revised version of the paper.

**170**

155

**List of all relevant changes made in the manuscript**

1. Title of the manuscript has been specified in a narrower sense.

2. The method of GHF estimation is modified: now we use the published depth-age scales at the studied sites and estimated the best value for the form factor.

175 3. We added tables with polynomial approximations of borehole temperature as a function of true vertical depth and "equivalent" thermophysical parameters.

4. Figure 1 is modified and location of the Antarctic ice divides is added.

5. Figure with comparison of the measured and modeled age scales with different form factors has been added.

180 6. The section with uncertancities evaluation has been added.

7. "Discussion" and "Conclusions" sections have been added.

On behalf of co-authors:

Pavel Talalay

185 Yazhou Li

**Geothermal heat flux from measured temperature profiles in deep ice boreholes in Antarctica**

Pavel Talalay1, Yazhou Li1, Laurent Augustin2, Gary D. Clow3, Jialin Hong1, Eric Lefebvre4, Alexey Markov1, Hideaki Matoyama5, Catherine Ritz4

[revised manuscript text omitted]
                    | W                        | AIS                        | EAIS                |                         |                          |                               |  |
|-------------------------------|--------------------------|----------------------------|---------------------|-------------------------|--------------------------|-------------------------------|--|
|                               | Byrd                     | WAIS                       | Vostok              | Dome C                  | Kohnen                   | Dome F                        |  |
|                               |                          | Divide                     |                     |                         |                          |                               |  |
| Coordinates                   | 80°01′ S,                | 79°28′ S,                  | 78°28′ S,           | 75°06′ S,               | 75° S,                   | 77°19′ S,                     |  |
|                               | 119°31′ W                | 112°05′ W                  | 106°48′ E           | 123°24′ E               | 0° E                     | 39°40' E                      |  |
| Years drilled                 | 1966-1968 a   | 2006-2011 d     | 1990–1998,          | 1999-2004 i  | 2002-2006 k   | 2003-2007 n        |  |
|                               |                          |                            | $2005-2014_{f,g}$   |                         |                          |                               |  |
| Surface elevation (m a.s.l.)  | 1530 a        | 1766 e          | $3488_{\rm f}$      | 3233 i       | 2892 1        | 3810 i             |  |
| Drilled depth (m)             | 2193                     | 3405 d          | 3769.3 g | 3270.2 i     | 2774.2 k      | 3035.2 n           |  |
| Ice thickness according with  | 2300 b        | 3455 e          | $3750\pm20_{g}$     | 3273±5 j     | 2750±501                 | 3028±15 j          |  |
| radar/seismic survey (m)      |                          |                            |                     |                         |                          |                               |  |
| Snow accumulation at surface  | 169.5c                   | 220 e           | 24.8 h   | 28.4j                   | 70 m          | 29.9 j             |  |
| (mm ice $a^{-1}$ )            |                          |                            |                     |                         |                          |                               |  |
| Ice sheet surface horizontal  | 12.7 o | ~3.0 t,u | 2.00±0.01s   | 0.015±0.01 p | 0.74 r | Negligiblev |  |
| velocity, m a -1   |                          |                            |                     | -                       |                          |                               |  |
| Mean surface snow temperature | -28 a         | -30 e           | -57 h    | -54.6 j      | -44 1         | -57.3 j            |  |
| (°C)                          |                          |                            |                     |                         |                          |                               |  |

aUeda, 2007; bWexler, 1961; cGow, 1968; dSlawny et al., 2014; eWAIS Divide Project Members, 2013; fVasiliev et al., 2011; gLukin and Vasiliev, 2014; hEkaykin et al., 2012; jAugustin et al., 2007; jParrenin et al., 2007a; kWilhelms et al., 2014; lUeltzhöffer et al., 2010; mHuybrechts et al., 2007; nMotoyama, 2007cWhillans, 1977; pVittuari et al., 2004; rWesche et al., 2007; sWendt et al., 2006; rConway and Rasmussen, 2009; uKoutnik et al., 2016; vMotoyama et al., 2008

I

| Table 2: Thermophysical properties at the base of Antarctic Ice Sheet at sites of deep ice-drilling estimated in this study |
|-----------------------------------------------------------------------------------------------------------------------------|
|-----------------------------------------------------------------------------------------------------------------------------|

| Parameters                                     | WAIS            |                   | EAIS                      |                 |                  |                   |
|------------------------------------------------|-----------------|-------------------|---------------------------|-----------------|------------------|-------------------|
|                                                | Byrd            | WAIS Divide       | Vostok                    | Dome C          | Kohnen           | Dome F            |
| Temperature, °C                                | -1.43           | -2.30             | -2.49                     | -2.15           | -1.85            | -1.99             |
| Temperature gradient (°C 100 m -1 ) | 3.70            | 3.88              | 2.02                      | 2.42            | 3.12             | 2.66              |
| Ice thickness according with depth             | 2164            | 3485              | 3759                      | 3257            | 2770             | 3016              |
| of pressure melting point (m)                  |                 |                   |                           |                 |                  |                   |
| Basal melt rate (mm a -1 )          | 1.2±0.8  | 3.7±1.7    | $-4.8\pm0.6$              | 1.08±0.27       | 2.8±1.6   | 2.5±0.5           |
| GHF (mW m -2 )                      | 88.4±7.6 | 113.3±16.9 | -3. 6 ±5. 3 | 57.9±6.4 | 86.9±16.6 | 7 8.9 ±5.0 |

Table 3: Polynomial approximations of borehole temperature T (°C) as a function of true vertical depth z and correlation factors

| Drill sites   | Polynomial                                                                                           | $\underline{R}^2$ |
|---------------|-------------------------------------------------------------------------------------------------------------|-------------------|
| Byrd          | $\underline{T} = -28.343 + 0.8367 \times 10^{-3} z - 6.7651 \times 10^{-6} z^2 + 6.1339 \times 10^{-9} z^3$ | 0.997      |
| WAIS Divide   | $\underline{T} = -31.799 + 8.8595 \times 10^{-3} z - 9.4649 \times 10^{-6} z^2 + 2.657 \times 10^{-9} z^3$  | 0.997      |
| Vostok | $\underline{T} = -56.034 + 2.9889 \times 10^{-3}z + 3.888 \times 10^{-6}z^2 + 0.2419 \times 10^{-9}z^3$     | 0.999      |
| Dome C        | $\underline{T} = -54.316 + 5.2978 \times 10^{-3}z + 4.4141 \times 10^{-6}z^2 - 0.368 \times 10^{-9}z^3$     | 0.999      |
| Kohnen        | $\underline{T} = -44.428 + 1.7384 \times 10^{-3} z + 4.4124 \times 10^{-6} z^2 + 0.184 \times 10^{-9} z^3$  | 0.999      |
| Dome F        | $\underline{T} = -55.016 + 5.839 \times 10^{-3} z + 5.188 \times 10^{-6} z^2 - 0.446 \times 10^{-9} z^3$    | 0.998             |

Table 4. GHF (mW m-2) calculated for different form factors m in the steady-state model, the correlation factor  $R^2$  between modeled and measured age scales and RSME

| Parameters                             | WAIS         |              | EAIS          |              |              |              |  |  |
|----------------------------------------|--------------|--------------|---------------|--------------|--------------|--------------|--|--|
|                                        | Byrd  | WAIS Divide  | Vostok | Dome C       | Kohnen       | Dome F       |  |  |
| $\underline{\text{GHF for } m = 0}$    | 8.72         | 72.9         | -16.16        | 134.6        | 66.04        | 92.78 |  |  |
| $\underline{R^2}$                      | -0.37 | -2.46        | 0.65          | 0.69         | 0.978 | 0.78         |  |  |
| RMSE                                   | 28.19        | 36.40        | 58.95         | 107.51       | 4.132        | 40.62        |  |  |
| $\mathbf{GHF \ for} \ m = 0.25$        | 28.44        | 100.0 | -35.28        | 105.3        | 60.95 | 130.9        |  |  |
| $\underline{R^2}$                      | 0.12  | 0.59  | 0.50          | 0.72         | 0.986 | -1.62        |  |  |
| RMSE                                   | 22.60        | 12.93 | 70.60         | 102.14       | 2.431 | 140.91       |  |  |
| $\mathbf{GHF \ for} \ m = 0.50$        | 55.05 | 207.4        | -20.33        | 70.06        | 156.4        | 92.96 |  |  |
| $\underline{R^2}$                      | 0.46         | 0.41         | 0.51          | 0.87         | 0.976        | 0.47         |  |  |
| RMSE                                   | 17.59 | 15.05 | 69.74  | 69.48 | 4.338        | 62.91 |  |  |
| $\underline{\text{GHF for } m = 0.75}$ | 95.84 | 240.3        | -25.39        | 57.40 | 106.0        | 104.8        |  |  |
| $\underline{R^2}$                      | 0.57  | 0.32         | 0.39          | 0.997 | 0.984        | 0.53         |  |  |
| RMSE                                   | 14.86 | 16.15 | 78.01         | 22.34        | 3.477 | 59.25 |  |  |
| $\underline{\text{GHF for } m = 1.00}$ | 117.8 | 251.3        | -8.90  | 67.3  | 161.5 | 79.20 |  |  |
| $\underline{R^2}$                      | 0.45  | 0.29         | 0.75   | 0.95  | 0.982        | 0.83  |  |  |
| RMSE                                   | 17.83 | 16.45 | 49.71  | 43.31 | 3.692 | 35.87 |  |  |

GHF values with the highest correlation factor and smallest RSME are highlighted by bold.

835 Table 5: Equivalent thermophysical parameters used by GA in comparison with published data

| Parameters                            | Byrd        | WAIS Divide  | Vostok | Dome C       | Kohnen       | Dome F       |
|---------------------------------------|-------------|--------------|---------------|--------------|--------------|--------------|
| "Equivalent" snow accumulation at     | 52.8 | 48.8         | 8.95   | 3.87         | 6.92  | 3.00         |
| surface (cm ice $a^{-1}$ )            |             |              |               |              |              |              |
| Modern snow accumulation at surface   | 16.9        | 22.0         | 2.48          | 2.84         | 7.00         | 2.99         |
| $(\text{cm ice } a^{-1})$             |             |              |               |              |              |              |
| "Equivalent" surface temperature (°C) | -29  | -30.9 | -56.5  | -54.7 | -44.9 | -56.5 |
| Modern surface temperature (°C)       | -28         | -30          | -57           | -54.6        | -44          | -57.3        |

---

## Author Response (AR2)

Dear Alex,

Thank you for your comments. We have addressed your concerns in the text and explanations are given below. Your comments are in brown, and our answers are in black.

I believe you have adequately addressed the reviewer comments. Please find several additional minor comments below, which should be addressed before publication.

In Section 2.2 where the equations are outlined, or directly in Section 2.3.2, I suggest adding the equation for the steady-state age profile as well. Since this is used to determine the best value for m further below, the equation should be stated.
We added equation as follows:
"Following the method of Fischer et al., (2013), we modeled the age profile of ice by

$$\hat{A} = \int_{z}^{H} \frac{1}{w_{melt} + (Acc - w_{melt})\left(\frac{z}{H}\right)^{m+1}} \, dz, \tag{9}$$

where $\hat{A}$ is the modeled ice age."

L23: Even the correlation factors => Even if the correlation factors
L24: we can use the GHF values => the GHF values estimated here … can be used as references
L27-28: is associated with sea level changes => can influence sea-level changes
L101: as well as => and
Corrected as recommended.

L109, Eq. 5: Please add a reference for this form of the equation, and particularly the form factor.
We added reference as follows: "According to Fischer et al. (2013), in the most general terms, the vertical velocity in the ice can be approximated by …"

L119: was designed => was used
L143: are coming from => come from
L257: there no => there is no
L257: drill site => the drill site
L373: we used GA approach => we used the GA approach
Corrected as recommended.

On behalf of the authors,
Pavel Talalay and Yazhou Li